# Chimera: A Multi-Task Recurrent Convolutional Neural Network for Forest Classification and Structural Estimation

**Tony Chang** [1,*,†] , **Brandon P. Rasmussen** [1,†], **Brett G. Dickson** [1,2,†] **and Luke J. Zachmann** [1,2,†]

1   Conservation Science Partners, Inc., 11050 Pioneer Trail, Suite 202, Truckee, CA 96161, USA;
    brandon@csp-inc.org (B.P.R.); brett@csp-inc.org (B.G.D.); luke@csp-inc.org (L.J.Z.)
2   Lab of Landscape Ecology and Conservation Biology, Landscape Conservation Initiative,
    Northern Arizona University, Box 5694, Flagstaff, AZ 86011, USA
*   Correspondence: tony@csp-inc.org; Tel.: +1-530-214-8905
†   These authors contributed equally to this work.

**Abstract:** More consistent and current estimates of forest land cover type and forest structural metrics are needed to guide national policies on forest management, carbon sequestration, and ecosystem health. In recent years, the increased availability of high-resolution ($<$30 m) imagery and advancements in machine learning algorithms have opened up a new opportunity to fuse multiple datasets of varying spatial, spectral, and temporal resolutions. Here, we present a new model, based on a deep learning architecture, that performs both classification and regression concurrently, thereby consolidating what was previously several independent tasks and models into one stream. The model, a multi-task recurrent convolutional neural network that we call the Chimera, integrates varying resolution, freely available aerial and satellite imagery, as well as relevant environmental factors (e.g., climate, terrain) to simultaneously classify five forest cover types ('conifer', 'deciduous', 'mixed', 'dead', 'none' (non-forest)) and to estimate four continuous forest structure metrics (above ground biomass, quadratic mean diameter, basal area, canopy cover). We demonstrate the performance of our approach by training an ensemble of Chimera models on 9967 georeferenced (true locations) Forest Inventory and Analysis field plots from the USDA Forest Service within California and Nevada. Classification diagnostics for the Chimera ensemble on an independent test set produces an overall average precision, recall, and F1-score of 0.92, 0.92, and 0.92. Class-wise F1-scores were high for 'none' (0.99) and 'conifer' (0.85) cover classes, and moderate for the 'mixed' (0.74) class samples. This demonstrates a strong ability to discriminate locations with and without trees. Regression diagnostics on the test set indicate very high accuracy for ensembled estimates of above ground biomass ($R^2 = 0.84$, RMSE = 37.28 Mg/ha), quadratic mean diameter ($R^2 = 0.81$, RMSE = 3.74 inches), basal area ($R^2 = 0.87$, RMSE = 25.88 ft$^2$/ac), and canopy cover ($R^2 = 0.89$, RMSE = 8.01 percent). Comparative analysis of the Chimera ensemble versus support vector machine and random forest approaches demonstrates increased performance over both methods. Future implementations of the Chimera ensemble on a distributed computing platform could provide continuous, annual estimates of forest structure for other forested landscapes at regional or national scales.

**Keywords:** remote sensing; recurrent convolutional neural networks; forest structure; forest classification; high resolution imagery; NAIP; multi-task learning

## 1. Introduction

Accurate estimates of above ground biomass (AGB) of vegetation in forests are fundamental for quantifying and monitoring forest conditions and trends. AGB and other forest structure metrics

provide baseline information required to derive estimates of available wood supply, habitat quality for wildlife, and fire threat, among other ecosystem attributes [1–4]. Such information can help guide, for example, national policies on forest management, carbon sequestration, and ecosystem health [5]. While in-situ measurements can provide a direct local-scale quantification of forest structure attributes, they can also be cost prohibitive, time consuming, and limited in spatial extent [6]. Therefore, utilization of widely available remotely sensed data with new modeling methods have become an essential approach for estimating AGB and other forest metrics in recent decades [7].

In recent years, advanced methods utilizing satellite or airborne sensor technology have resulted in regional and global-scale estimates of forest structure [8–12]. These sensors range from very high spatial resolution optical/laser datasets (Quickbird and airborne LiDAR) to moderate or low spatial resolution datasets (Landsat, radar, and MODIS), each with their own advantages and disadvantages [7,13]. Airborne LiDAR is currently the most advanced platform and has recently become ubiquitous in the literature. This is due to the high accuracy in forest structure metrics estimation produced by its very high spatial resolution (sub 1-m), and its ability to represent three-dimensional structure with point clouds [14]. However, despite its growing usage, airborne LiDAR acquisitions are costly, limited in spatial and temporal coverage, and can contain fly-over data gaps, thus limiting their utility for large continuous regional analyses of forest structure [7].

Landsat has been the most widely used sensor to measure forest structure due to its free availability, medium temporal resolution (16-day return interval), and effective global spatial coverage [15]. The wall-to-wall access of data globally has led to advancements in understanding not only of the patterns and dynamics of AGB, but also of net primary productivity and forest canopy cover change dynamics [9,16]. Yet, its moderate spatial resolution (30-m) has led to issues of "data saturation," in which a single pixel's reflectance has high uncertainty and/or underestimates at AGB values above 150 Mg/ha [1,17,18]. This is due to mature forests often containing a complex mixed age structure, resulting in canopy layering, which can be difficult to discern at a 30-m spatial resolution [19]. One data source that has the potential to resolve issues with Landsat's data saturation and LiDAR's limited coverage is the USDA National Agriculture Imagery Program (NAIP) [20]. Since 2002, the NAIP program has provided freely accessible, 1-m resolution aerial photography, approximately every two years and for each state in the conterminous US [20]. Using texture/pattern recognition software, such high resolution imagery can help to overcome data saturation issues and provide better estimates of forest structure metrics [21,22].

In the past couple of decades, development of newer machine learning methods, including deep learning, have led to an explosion of computer vision research, lending itself to the creation of complex and computationally efficient relationship models in the field of remote sensing [23–29]. One application for which deep learning is particularly well-suited is image interpretation and pattern recognition with spatial data through use of convolutional neural networks (CNN) [30]. CNNs have proven to outperform all other existing methods when applied to these tasks [31–33]. Yet, this method has not been readily adopted for forest metric estimation, such as AGB [1]. Predictions of forest structure metrics have been shown to improve when including image textures as predictors, e.g., gray level co-occurrence matrix (GLCM), standard deviation of gray levels (SDGL). However, these, and other more complex textural features require laborious hand-crafting to identify and implement in model fitting [34–36]. CNNs have the ability to identify similar relevant image textures from the data alone without human assistance, which allows them to be applied to solve generalized image feature detection problems [32,37]. Additionally, the advancement of recurrent CNNs (RCNN), convolutional neural networks that include a time dependent layer, allows identified image texture spaces to be ordered in sequence [38,39]. The ability of RCNNs to recognize patterns in multi-dimensional domains that have both time and space components demonstrates promise with object recognition in a wide range of remote sensing problems, from classification to segmentation [40–42].

With the rapid adoption of CNNs for solving computer vision problems, additional advanced approaches have been developed to improve predictive accuracy, which include multi-task learning

and ensembling. Multi-task learning is a method in which a neural network is trained to perform two or more similar tasks, such as identifying road characteristics while predicting steering direction for an automated driving system [43]. This shared task method of learning has been demonstrated to increase predictive ability for each individual task [44]. Ensembling, the technique of using multiple models together to make a single prediction, has also been demonstrated to increase predictive ability of many machine learning approaches, by reducing the bias of any single model prediction [45]. These technical advancements and characteristics present an opportunity to apply them in an ensemble of multi-task RCNNs as a case study, and test the ability of such an architecture to classify forest cover and predict AGB and other forest structure metrics against more conventional approaches.

In this study, we introduce an ensemble of individual RCNN models called "Chimera" (together called the Chimera ensemble, or CE) to perform a data fusion of high resolution NAIP imagery, moderate resolution time-varying Landsat, and ancillary climate and terrain (ANC) variables, and to build prediction tiles which can then be reassembled for spatially explicit mapping of larger areas. We present performance metrics based on field plots collected from the USDA forest service forest inventory and analysis (FIA) program dataset.

*Objectives*

The main objective of this study is to measure the performance of a novel multi-task RCNN architecture which simultaneously classifies forest and land cover ('conifer', 'deciduous', 'mixed', 'dead', or 'none') and estimates forest structure metrics (AGB, quadratic mean diameter (QMD), basal area, and canopy cover). Our performance experiments will involve: (1) comparing the different combinations of input datasets, specifically, the impact of including high resolution (1-m) imagery; (2) comparing the RCNN architecture with more commonly used random forest (RF) and support vector machine (SVM) models, with similar inputs; (3) assessing the potential improvement with ensembling of RCNN models compared to an individual best fit model. It is hoped that these objectives (Figure 1) will better inform the application of deep learning approaches in forest structure and type estimation, and present a potential architecture upon which future modeling efforts might improve.

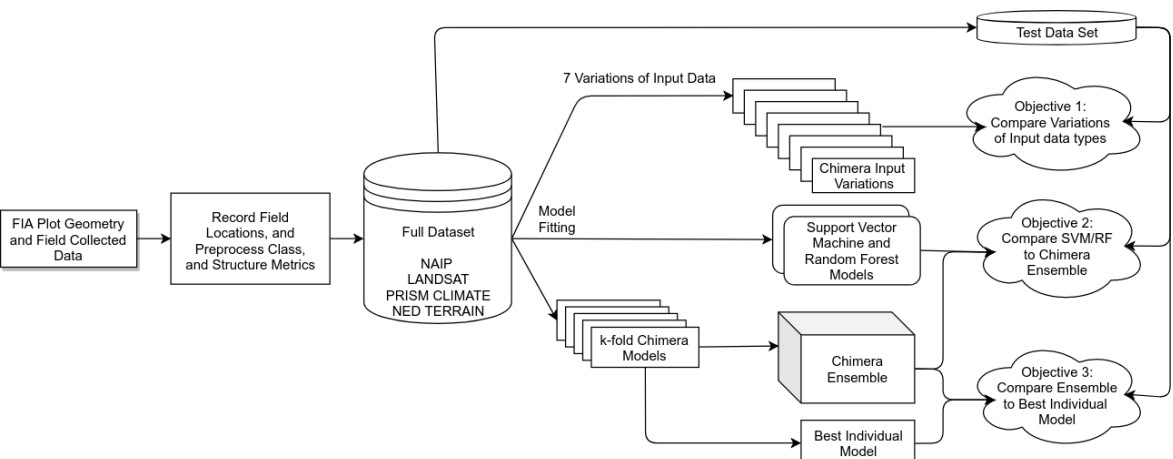

**Figure 1.** Flowchart of study workflow and major objectives.

## 2. Materials and Methods

### 2.1. Region of Analysis

California and Nevada were chosen as the region of analysis for data sampling and modeling (Figure 2). The California–Nevada region embodies a wide gradient of elevation from −86 m in Death Valley to 4421 m at the summit of Mt. Whitney. In California, about 40% of the area (approx. 13.35 million hectares) is covered by forests [46], while Nevada contains the largest national forest in

the lower 48 states, the 6.3-million acre Humboldt–Toiyabe National Forest [47]. Both states contain a wide variety of ecosystems including alpine, montane and subalpine forests, coastal forests, mixed conifer-deciduous forests, chaparral, pinyon-juniper woodlands, and desert scrub. This variety of forest ecotypes make these states an excellent case study for forest structure estimation over an extensive, heterogeneous region.

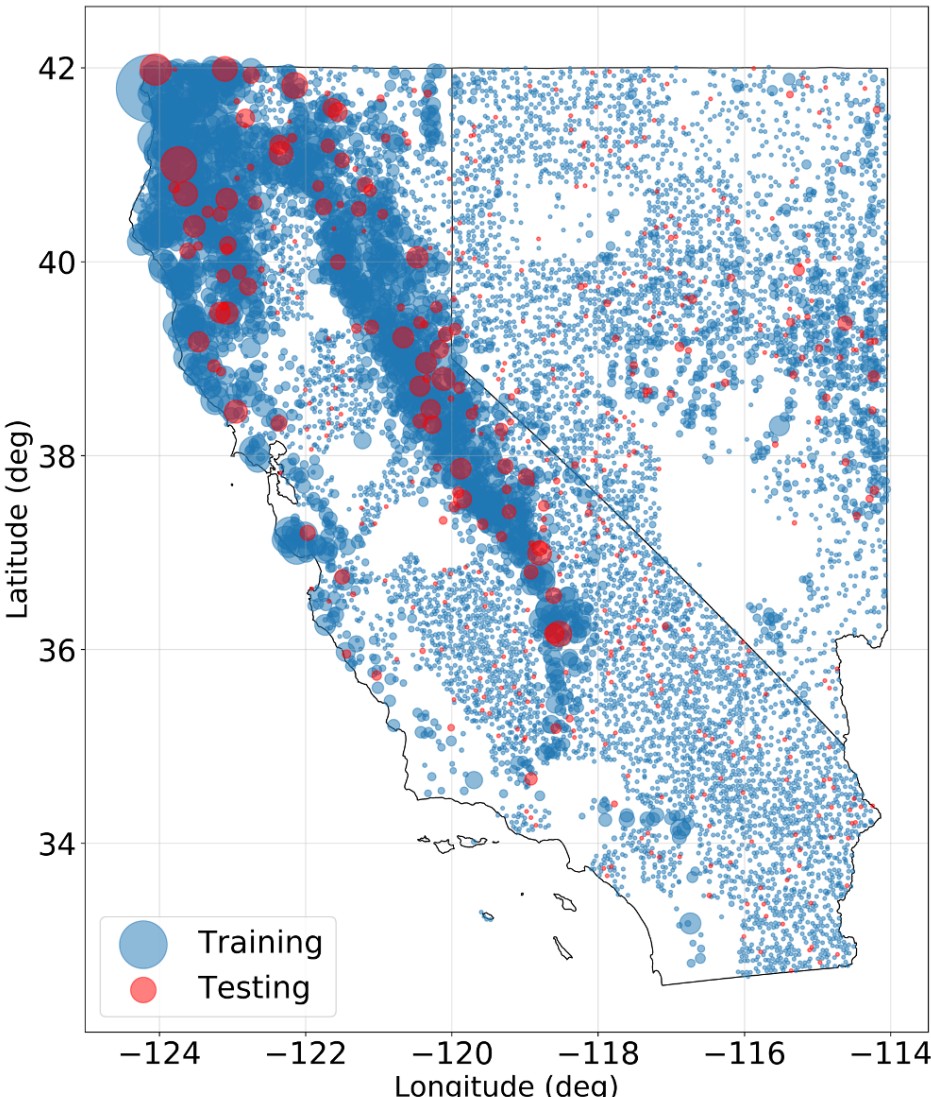

**Figure 2.** Case study region of California and Nevada for training and testing the Chimera model. The region of interest provides high variability of forested and non-forested cover types. Image variability creates a range of challenges for estimation of forest structure metrics. Training (blue) and testing (red) Forest Inventory and Analysis (FIA) plot accurate estimates of above ground biomass (AGB)'s are displayed here as scaled dots (minimum = 0 Mg/ha, maximum = 3240.2 Mg/ha) to demonstrate range of spatial variability within the study region (locations here are approximate to maintain FIA spatial confidentiality).

*2.2. Predictor Variables—Optical Data*

Our input variables for the RCNN architecture are formed from a combination of four different, open-source remote sensing datasets available in the Google Earth Engine (GEE) [48]. These datasets (NAIP imagery, LANDSAT imagery, PRISM climate data, and USGS NED terrain data) are combined across different spatial and temporal resolutions to form individual tensors (i.e., multi-dimensional array), each approximating a single 120 × 120-m FIA field measured sample for each of our predictor

variables (Figure A3). We chose the 120 × 120-m size to maximize the inclusion of the sampled area while allowing for any potential mismatches in either imagery georeferencing, or FIA plot location [49].

### 2.2.1. NAIP (National Agricultural Imagery Program)

USDA NAIP images are mostly cloud-free image tiles collected across the continental United States during the summer months by aerial fly-over in frequencies ranging from 5 years (in some rural areas) to 1 year [20]. NAIP image tiles are sized at 3.75 × 3.75 arc-minutes (approx. 6 km × 6 km) in GEE, with either three (red, green, blue; RGB) or four (red, green, blue, and near-infrared starting in 2007; RGBN) spectral bands, with a target of 1 m or smaller pixel size. NAIP is available across the entire time range of FIA inventory samples that we considered (2005–2017). In order to limit the cases of non-representative samples, locations without NAIP imagery available within a three year time window (one year forward, two years back from the date of FIA sampling) were not included in the training dataset. The image collected closest to the date of FIA field data collection was chosen and resampled to 1 meter resolution either through mean aggregation, or nearest neighbor resampling. Only RGB bands were used due to lack of uniform availability of RGBN images across the two states in our case study region.

### 2.2.2. Landsat 7

Landsat 7 is one of a series of satellites providing open-source imagery for scientific purposes through a joint effort by the USGS and NASA [15]. Landsat 7 imagery is collected at 16-day intervals through the entire period of interest for this study. For training purposes, we produced a 4 × 4-pixel image at the native 30-m resolution of all but the thermal bands, resampling the panchromatic band to 30 m from 15 m via mean aggregation, and utilizing the quality assessment (QA) band codes to mask noisy pixels and all but the lowest confidence cloudy pixels. This was accomplished using only tier 1, top-of-atmosphere (TOA) reflectance data. As a result of gaps in surface reflectance data and reduced efficacy of data correction over arid/snow covered regions in GEE, we used TOA to maintain the highest possible number of continuous temporal samples [50]. We gathered a three year time series (one year forward, two years back from FIA sampling) and produced a monthly average for each of the 12 calendar months of reflectance values with only high quality pixels to represent a single year. This method provided the best chance at capturing the distinct patterns produced by deciduous and mixed stands, as well as dead stands through an annual cycle.

### *2.3. Predictor Variables—Ancillary Data*

Kane et al. [51] were able to demonstrate that forest structure patterns such as canopy cover percentage, could be predicted based on water balance and topographic position information. This follows the intuition that specific forest species with unique structural attributes can be delimited based on abiotic factors [52]. Therefore, we included ancillary (ANC) climate and terrain data to provide the CE with additional information for land cover classification and forest structure estimation.

### 2.3.1. PRISM (Parameter-Elevation Regressions on Independent Slopes Model)

In order to provide long term climate trend information to the CE for each sample, we utilized PRISM from Oregon State University [53]. PRISM aggregates continuous data from weather stations across the United States and uses sophisticated interpolation methods to produce gridded data for the entire United States on multiple time scales. We utilize AN81m, a monthly curated and quality-controlled dataset which provides dewpoint, vapor pressure deficit, temperature, and precipitation data at a coarse 2.5 arc-minute (approx. 4 km) scale. This dataset was resampled to a single 120-m pixel for each sample using the nearest neighbor method, and a mean/standard deviation across the 30-year period prior to the FIA inventory year associated with each training or test sample.

### 2.3.2. USGS NED (National Elevation Dataset)

The National Elevation Dataset (NED) provides gridded elevation at a 1/3 arc-second (approx. 10–15 m) resolution for the entire continental U.S. [54]. NED was produced from multiple individual datasets as a national aggregation and has been shown to have an RMSE $\approx$ 1.63 m from truth in vertical accuracy across the continental U.S. [55]. Terrain features including elevation, slope, and aspect, have been shown to be correlated with forest structural attributes [56,57]. We use elevation, slope, and aspect (which we calculate and apply a cosine transform on the fly directly from the NED) in the training process, represented as a single, 3-band, 120 m mean aggregated pixel for each sample.

### 2.4. Response Variables

Training Data Targets: FIA Sampling and Database Parameter Usage

Continuous response variables were generated from the forested and non-forested 27,966 FIA Phase 2 plots inventoried from 2005–2017 within California and Nevada state boundaries. The FIA program applies a nationally consistent quasi-systematic sampling protocol [58] with a nominal sampling intensity of approximately one sample location per 2400 ha, to provide a source of information about the extent, condition, status, and trends of forests across the United States [8,59]. In Phase 2, field crews visited permanent plots that contain a forested land use (areas that have at least 10% tree canopy cover, are at least 0.4 ha in size, and at least 36.6 m wide) and collect information on individual trees and site variables [60]. In each plot, trees were sampled at the subplot (four 24 foot (7.31 m) diameter circles) or macroplot (58.9 foot (17.95 m)) level in a consistent orientation. Only larger trees are measured outside the subplots in the case of macroplot measurements by FIA field crews. We considered all live trees above 1 inch (2.54 cm) in diameter when calculating our response variables. Dry biomass components for each live tree were calculated via the national standard method (the component ratio method (CRM)) detailed in the current FIA handbook [60]. We summed the total above ground components of each tree together to represent the dry biomass of a sample plot, accounting for the variations of the CRM for woodland species, timber species, and saplings. These values were then converted to a density value using recorded plot geometry information. Basal area metrics were taken directly from FIA field measurements within the database. Canopy cover measurements were taken from the live canopy cover attribute of the database, which can be measured differently (in-situ or remotely sensed image interpretation) depending on the region and year of survey [61]. Plot-level QMD was given by the equation:

$$QMD = \sqrt{\frac{\sum dbh_i{}^2}{n}} \tag{1}$$

where *dbh* is the diameter at breast height of each tree, $i$, for all measured live trees, $i = 1, ..., n$ on the plot.

For the land cover classification task of this study, we classified plots based on an 80% tree count threshold as 'conifer' or 'deciduous'. If there was not 80% trees of conifer or deciduous type, plots were labeled as 'mixed'. A label of 'dead' was given to plots where 80% of the trees were recorded as standing dead. Plots where no trees were present were labeled as 'none'. It should be noted that these classification labels are different from FIA definitions, which are land-use labels and not land cover labels. We additionally utilized FIA non-forested plots (20,660 plots of the 27,966). These plots provide a valuable baseline for image feature recognition of urban areas, bare land, and water. However, since these plots are non-forested by definition, these locations are not-visited and have no tree attributes recorded but may still have tree cover (e.g., urban and residential areas) [61]. Following the methods of Hogland et al. [35], who visually inspected plot locations against the corresponding reference NAIP image, we manually inspected images corresponding to these non-visited plots. For locations where there were clearly no trees visible in the imagery, we labeled them as 'none' and attributed them with AGB, QMD, basal area, and canopy cover values of zero.

After all manual inspections of plot-to-image correspondence were completed, plots were further filtered to those containing all four corresponding predictor variable data, leaving *n* = 9967 samples (Table 1). The data were then randomly subdivided into training, validation, and test samples. We first set aside 500 plots for testing (5% of the total data) and then used the remaining data at a 4:1 ratio of training to validation data for fitting each individual Chimera model (7724 training samples, 1743 withheld validation samples, and 500 test samples). Training, validation, and test samples followed similar distributions for each response metric, and across all *k*-fold cross-validation subsets as discussed in the *Model Ensembling* section below (Figure A1).

**Table 1.** Summary of forest inventory and analysis (FIA) plot data: (**a**) distribution of FIA plot forest type classes for all plots used in training/validation and testing; (**b**) bulk statistics of FIA plot forest structure metrics (above ground biomass (AGB), quadratic mean diameter (QMD), basal area, and canopy cover) for all forested plots used in training (*n* = 3237) and testing (*n* = 170).

**a**

| Class | Training/Validation | | Testing | |
|---|---|---|---|---|
| | *n* | (%) | *n* | (%) |
| None | 6226 | 0.66 | 330 | 0.66 |
| Conifer | 1686 | 0.18 | 92 | 0.18 |
| Deciduous | 407 | 0.04 | 15 | 0.03 |
| Mixed | 1105 | 0.12 | 62 | 0.12 |
| Dead | 43 | $<< 1$ | 1 | $<< 1$ |

**b**

| Metric | Training/Validation | | | | | Testing | | | | |
|---|---|---|---|---|---|---|---|---|---|---|
| | Mean | SD | Min | Median | Max | Mean | SD | Min | Median | Max |
| AGB (Mg/ha) | 99.62 | 139.04 | 0.1 | 41.0 | 3240.2 | 103.39 | 134.48 | 0.1 | 40.8 | 907.4 |
| QMD (in) | 15.15 | 7.20 | 1.0 | 13.2 | 87.2 | 15.61 | 7.49 | 5.0 | 13.4 | 42.1 |
| Basal Area (ft$^2$/ac) | 107.69 | 88.14 | 0.6 | 88.1 | 1201.6 | 107.84 | 85.48 | 1.1 | 86.6 | 435.2 |
| Canopy Cover (%) | 40.56 | 25.76 | 0.5 | 35.0 | 99.0 | 40.21 | 24.71 | 3.0 | 35.1 | 99.0 |

*2.5. Chimera RCNN Architecture*

We constructed a deep learning model architecture called Chimera, for estimating forest structure using the Keras (v2.1.3) package [62] for building TensorFlow (v1.3.0) CNNs in Python. We implemented a multi-task learning (MTL) neural network architecture that merges climate, terrain, Landsat, and NAIP imagery datasets as predictors to perform two tasks: (1) classification of land cover type ('none' (non-forested), 'deciduous', 'conifer', 'mixed', 'dead') and (2) regression of four continuous forest metrics (AGB, quadratic mean diameter (QMD), basal area, and canopy cover) simultaneously [44] (Figure 3). Multi-task learning is a technique that has been demonstrated to improve accuracy by allowing convolutions to fit parameters more efficiently through understanding "easier" tasks [43,63,64]. This follows the intuition from another popular model called "you only look once" (YOLO) [65], that performs both object detection and classification simultaneously. In the YOLO architecture, a single trained CNN takes an image as an input and then performs discrete object identification (classification task) and draws a box around the object, thus performing an estimate of the box width, height, and center coordinates (regression tasks). Our architecture, similarly performs two tasks, but also allows the neural network to learn multiple differing input feature parameters simultaneously. To perform the classification and regression tasks, Chimera minimizes two loss functions in a single model; a cross-entropy loss function (*H*):

$$H(\boldsymbol{y}, \hat{\boldsymbol{y}}) = -\sum_i y_i \log(\hat{y}_i), \tag{2}$$

where $y_i$ is the ground truth label of $i$th training sample instance and $\hat{y}_i$ is the $i$th model prediction, and the L2 (mean squared error; MSE) loss function:

$$\text{MSE}(\boldsymbol{y}, \hat{\boldsymbol{y}}) = \frac{1}{n} \sum_{i}^{n} (\hat{y}_i - y_i)^2. \tag{3}$$

Chimera's architecture utilized a five block DenseNet structure to serve as the shared layer that will identify features within a three channel NAIP image. The DenseNet architecture [66] allows improved information flow between sequential convolutional composite layers $F_j$ composed of a batch normalization, rectified linear units (ReLU), convolution, dropout, and pooling. We denoted the output of the $j$-th layer as $x_j$ by direct concatenation of the feature maps produced from proceeding layers:

$$x_j = F_j([x_0, x_j, \ldots, x_{(j-1)})], \tag{4}$$

where $[x_0, x_j, \ldots, x_{(j-1)}]$ refers to the concatenation of the $j$ previous layer outputs. This improved information flow additionally allows for higher performance, reduced parameter space, and faster model fitting compared to other well known architectures (e.g., ResNet, GoogLeNet, AlexNet) [32,37,67].

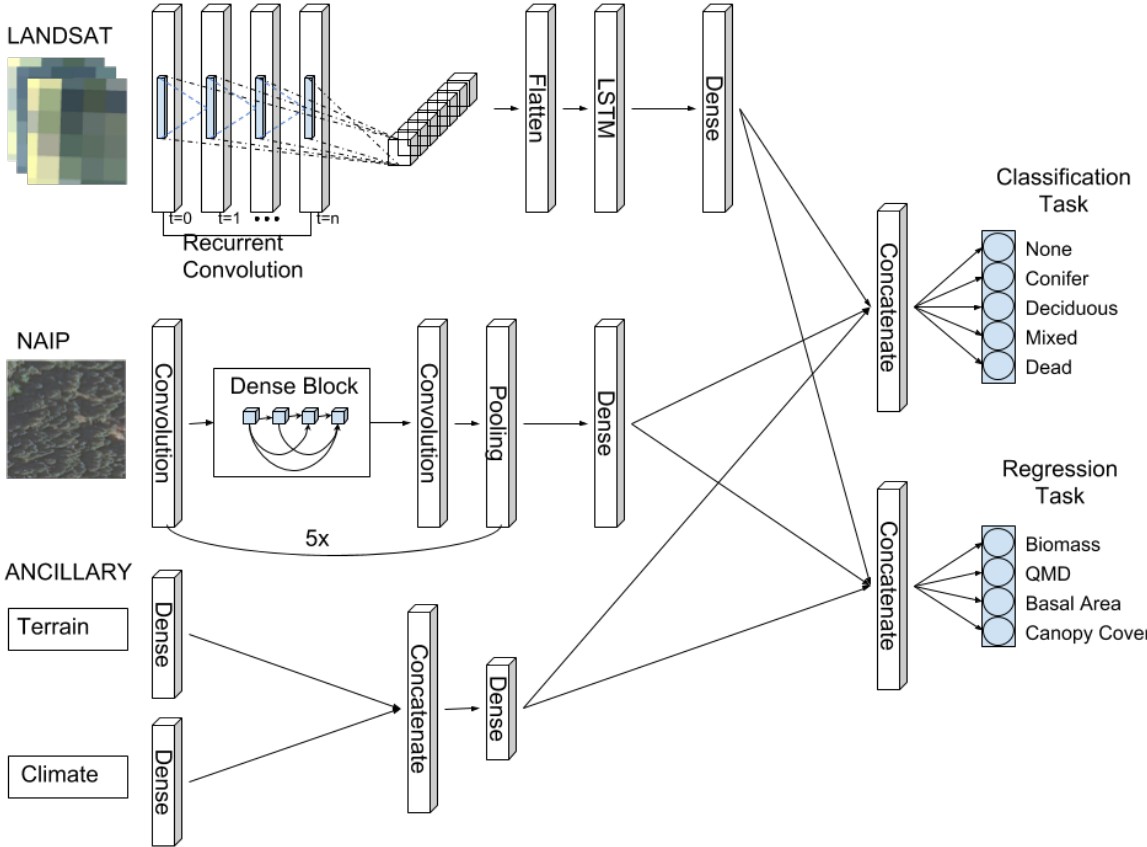

**Figure 3.** The recurrent convolutional neural network architecture used by the Chimera multi-task network for forest structure regression and forest type classification. Chimera takes in a single National Agriculture Imagery Program (NAIP) aerial image, a three year time-series of Landsat aggregated to the month, and ancillary terrain/climate measurements of varying resolutions and combines features from each input for estimation.

A three layer convolutional long short-term memory (LSTM) block was applied to time sequenced Landsat scenes to summarize temporal changes that would be conducive to identifying deciduous, conifer, and mixed forest stands [38,68,69]. The output of the LSTM layer was concatenated to the dense layer of the DenseNet for classification.

We concatenated the ancillary climate and DEM data to the output of the final DenseNet layer and LSTM layer. We then forked the concatenated layer in two for each task and run each fork through four fully connected layers with five outputs for the classification task and four outputs for the regression task. This is an example of a hard parameter sharing architecture, in which the same final concatenated layer is used for both tasks [44]. This allows the classification loss to influence regression parameters, and the regression loss to influence the classification parameters. For example, where classification was 'none', forest structure metrics were low to zero, indicating a strong relationship between the two tasks.

NAIP and Landsat image samples were augmented randomly using image rotation and mirroring during model training to increase robustness of the RCNN and to allow varying forest image feature geometries to represent the same forest structure metric values across different training epochs. Furthermore, to compensate for training sample class count imbalance, we applied the Eigen and Fergus [70] class weighting method to adjust the final cross-entropy loss layer.

Each individual Chimera RCNN was trained for 90 epochs at an 80 sample batch size using a stepped learning rate Adam optimizer [71] (learning rate stepped by a factor of 0.1 at every 30 epochs) to increase speed of model convergence during gradient descent. Average training time was 1h30m on a Microsoft Azure NC6 instance using a Intel (R) Xeon (R) E5-2690v3@2.80GHz CPU with 56 GB of RAM and a NVIDIA Tesla K80 GPU with 24 GB of GDDR5 memory.

Additionally, to understand if contributions of each input dataset improve performance, we fit seven different independent models with the complete training dataset, using all possible combinations of input data with the Chimera architecture (ANC only; NAIP only; NAIP and ANC; Landsat only; Landsat and ANC; NAIP and Landsat; all inputs).

## 2.6. Chimera Comparisons with RF and SVM

Using the same training and test data sets, we built inputs for random forest and support vector machine models with the Scikit-learn package implemented in Python to compare results [72]. Data were subsetted to accomplish each regression task (four models), and the classification task separately (one model), resulting in five models for each algorithm. Two forms of data aggregation were tested: one used a single Landsat pixel time series of aggregated arrays resulting in 84 features, while the other used the same full $4 \times 4$ Landsat image time series as the Chimera ensemble, resulting in 1344 features. In both cases, textural extraction of NAIP was performed identically: one layer of $4 \times 4$ standard deviation gray level (SDGL) and one layer of $4 \times 4$ mean-downsampled images were concatenated, resulting in 96 features. Climate and DEM inputs were kept in the same shape and all data were normalized through the same process as the Chimera Ensemble. The final shapes of the resultant flattened vectors were 197 and 1457 features. The RF models were parameterized with $n_{estimators} = 150$ and the SVM models were parameterized with $c = 250, \gamma = 0.5$ after employing a grid search methodology and finding the best output RMSE values across all tasks.

## 2.7. Model Ensembling

We used a *k*-fold cross-validation methodology to divide the data into a 4:1 ratio training versus validation samples. Five separate Chimera models ($k = 5$) were fit with each fold of the training data. We performed diagnostics on each model to ensure all had converged on their loss functions and accuracy metrics (Figure 4). A super learner [45] approach was used to ensemble the five models weighted by the design matrix $A$ in the form:

$$\hat{Y}_{ensemble} = A\hat{Y}_{stack}, \tag{5}$$

where $\hat{\boldsymbol{Y}}_{stack}$ is a matrix of stacked column vectors $\hat{\boldsymbol{Y}}_k$, for each fitted model $k$. We solved for $\boldsymbol{A}$ using a multiple linear regression as the meta-learning algorithm and applied these final weights to combine models for prediction. The stacking methodology has been demonstrated to increase prediction accuracy metrics over a single model alone, or a simple unweighted averaging of models [73].

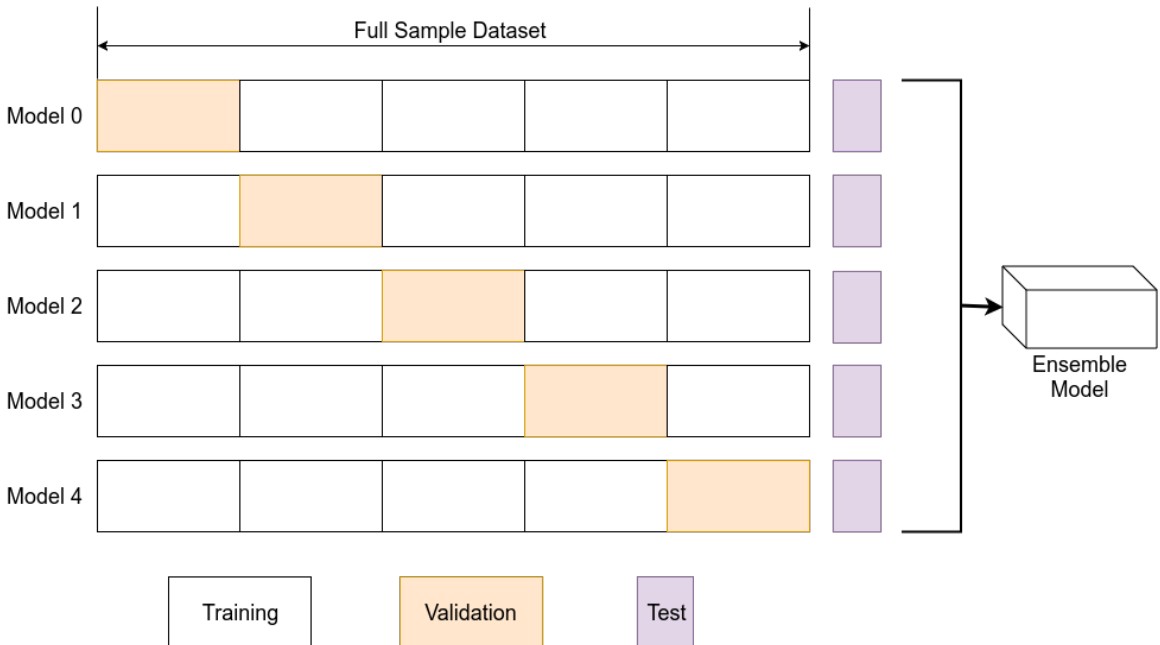

**Figure 4.** Diagram of the *k*-fold cross validation and model ensemble stacking for model diagnostics and prediction. All models utilized a total of 9967 samples divided into: 7724 training samples and 1743 *k*-fold withheld validation samples, both unique to each model. We also generated 500 independent test samples for final evaluation. Model ensembling was performed using a super learner model stacking methodology [45].

## 3. Results

### 3.1. Model Diagnostics and Input Experiments

Both the batch-level classification and regression loss functions converged for training and validation data for all *k*-folds within 60 epochs for a total of 1,353,593 parameters. Low differences in loss values between training and validation suggested that no individual Chimera model was overfit (Figure 5). Experiments of differing Chimera model input combinations (Table 2) demonstrated that all data types on their own (ANC only, NAIP only, Landsat only), did not perform as well as models with two input types. NAIP + ANC data outperformed Landsat models in classification (+0.01 overall F1-score) and similarly to Landsat + ANC in regression performance ($R^2$ = 0.76). The full model case (NAIP + Landsat + ANC) had the best overall performance (overall classification F1-score = 0.90; overall regression $R^2$ = 0.81, normalized RMSE = 0.076) for the combined task of classification and regression simultaneously compared to each input combination considered separately, however the imagery-only model (NAIP + Landsat) performed slightly better in land use cover plot classification (overall classification F1-score = 0.92; overall regression $R^2$ = 0.78, normalized RMSE = 0.080).

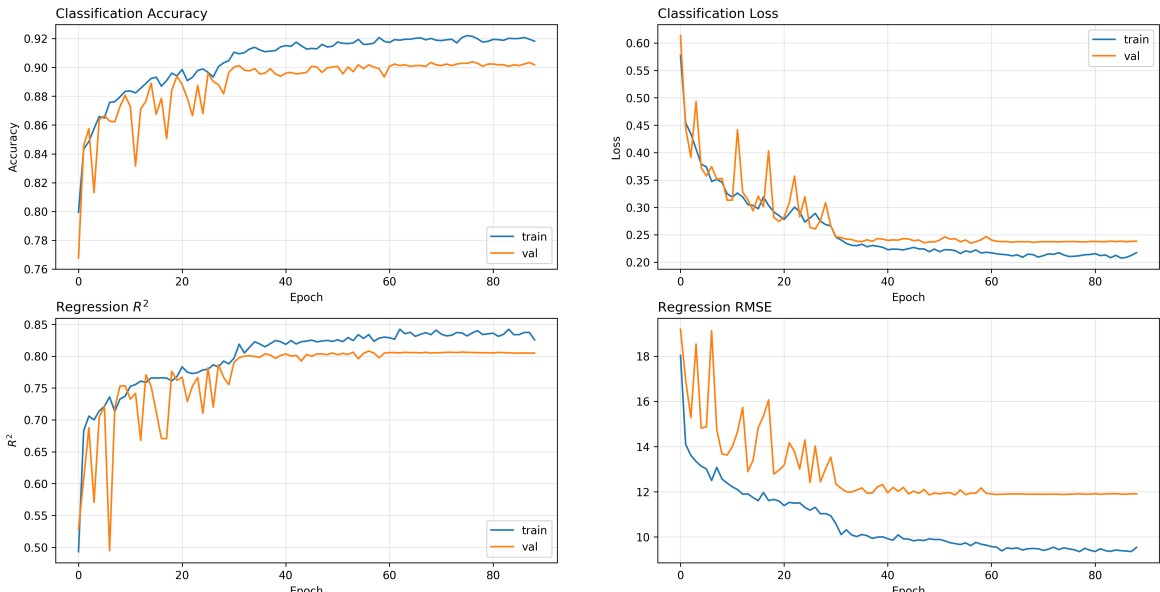

**Figure 5.** A single Chimera model fit history for both classification and regression tasks using a cross-entropy loss function (classification) and a L2 loss function (regression). All models had training losses approximately converge by the 60th epoch.

**Table 2.** Experiment results of varying combinations of input training data types (ancillary (ANC) only; National Agriculture Imagery Program (NAIP) only; NAIP + ANC; Landsat (LS) only; LS + ANC; NAIP + LS; NAIP + LS + ANC (FULL)) for individual Chimera models. Full composite of data inputs results in best overall performance on a single Chimera model for both classification and regression simultaneously. All statistics reported here are for the training-withheld test set of 500 plots. Best scores (and best score ties) are denoted in bold.

| Classification Metric (F1-Score) | ANC | NAIP | NAIP + ANC | LS | LS + ANC | NAIP + LS | FULL |
|---|---|---|---|---|---|---|---|
| None | 0.83 | **0.99** | **0.99** | 0.97 | 0.97 | **0.99** | 0.98 |
| Conifer | 0.57 | 0.80 | 0.82 | 0.78 | 0.81 | **0.85** | 0.81 |
| Deciduous | 0.00 | 0.46 | 0.39 | 0.27 | 0.39 | **0.65** | 0.60 |
| Mixed | 0.27 | 0.59 | 0.61 | 0.56 | 0.63 | **0.74** | 0.66 |
| Dead | 0.00 | 0.00 | 0.00 | 0.00 | 0.00 | 0.00 | 0.00 |
| Overall | 0.68 | 0.89 | 0.89 | 0.86 | 0.88 | **0.92** | 0.90 |
| **Regression Metric ($R^2$)** | **ANC** | **NAIP** | **NAIP + ANC** | **LS** | **LS + ANC** | **NAIP + LS** | **FULL** |
| AGB | 0.00 | 0.70 | 0.70 | 0.74 | 0.73 | 0.75 | **0.78** |
| QMD | 0.12 | 0.78 | 0.76 | 0.68 | 0.67 | 0.78 | **0.79** |
| Basal Area | 0.04 | 0.75 | 0.76 | 0.78 | 0.79 | 0.78 | **0.81** |
| Canopy Cover | 0.07 | 0.82 | 0.82 | 0.83 | 0.83 | 0.84 | **0.85** |
| Overall (mean $R^2$) | 0.06 | 0.76 | 0.76 | 0.76 | 0.75 | 0.78 | **0.81** |
| **Regression Metric (RMSE)** | **ANC** | **NAIP** | **NAIP + ANC** | **LS** | **LS + ANC** | **NAIP + LS** | **FULL** |
| AGB (Mg/ha) | 92.62 | 50.67 | 50.45 | 46.95 | 48.05 | 46.45 | **43.34** |
| QMD (in) | 8.07 | 3.99 | 4.24 | 4.87 | 4.90 | 4.04 | **3.90** |
| Basal Area (ft$^2$/ac) | 69.90 | 35.86 | 34.65 | 33.16 | 32.72 | 33.42 | **30.91** |
| Canopy Cover (%) | 23.05 | 10.04 | 9.99 | 9.85 | 9.99 | 9.47 | **9.25** |

### 3.1.1. Classification Task Diagnostics

We used accuracy metrics of precision, recall, and F1-scores to assess Chimera's classification abilities. Precision refers to number of true positives divided by total true and false positives, while recall refers to true positives divided by the sum of true positives and false negatives. The F1-Score is a combination of the precision and recall, defined by, $(2 * \text{precision} * \text{recall})/(\text{precision} + \text{recall})$. Classification diagnostics on the test set reported an overall average precision, recall, and F1-score of

0.92, 0.92, and 0.92, respectively (Table 3) (Figure 6). Class-wise F1-scores were high for 'none' (0.99) and 'conifer' (0.86) classes. This indicated that the ensemble model was able to distinguish between forested and non-forested plots. The CE had difficulty accurately distinguishing the 'deciduous' class and 'mixed' class, with moderate F1-scores of 0.6 and 0.74 respectively. This difficulty emerged as a result of class label imbalance in the training data. The 'dead' class samples were insufficiently assessed due to the low sample number within the test samples ($n = 1$). Due to this limited number of dead samples in the test set, we assessed each individual k-fold Chimera model in CE for their predictive ability on their respective k-fold validation subset. F1-scores for these k-fold validations were as follows: ($k_1 = 0.222$, $k_2 = 0.182$, $k_3 = 0.0$, $k_4 = 0.0$, $k_5 = 0.133$), demonstrating a weak predictive ability for the dead class. Comparisons to SVM and RF reported higher levels of performance from the CE in all classes (except 'dead' class), with the largest difference occurring with the CE versus the SVM in classifying the 'deciduous' (F1-score difference of 0.26) and 'mixed classes' (F1-score difference of 0.13) (Table 3).

**Table 3.** Classification accuracy metrics for the best individual Chimera model (BIM), Chimera ensemble (CE), random forest (RF), and support vector machine (SVM) used in this study to predict forest type from FIA plot data. All statistics reported here are for the training-withheld test set of 500 plots. Precision refers to number of true positives divided by total true and false positives, while Recall refers to true positives divided by the sum of true positives and false negatives. The F1-Score is two times the precision times the recall, divided by the sum of precision and recall. Support refers to the number of samples for each class. Best scores (and best score ties) are denoted in bold.

| Class | Precision | | | | Recall | | | | F1-Score | | | | Support |
|---|---|---|---|---|---|---|---|---|---|---|---|---|---|
| | CE | BIM | SVM | RF | CE | BIM | SVM | RF | CE | BIM | SVM | RF | |
| None | **0.99** | 0.98 | 0.96 | 0.97 | **0.99** | **0.99** | 0.97 | **0.99** | **0.99** | **0.99** | 0.96 | 0.98 | 330 |
| Conifer | 0.81 | **0.82** | 0.73 | 0.79 | **0.91** | 0.90 | 0.76 | 0.87 | **0.86** | **0.86** | 0.74 | 0.83 | 92 |
| Deciduous | **0.60** | 0.54 | 0.36 | 0.54 | **0.60** | 0.47 | 0.33 | 0.40 | **0.60** | 0.50 | 0.34 | 0.46 | 15 |
| Mixed | **0.81** | **0.81** | 0.63 | 0.78 | **0.68** | **0.68** | 0.58 | 0.65 | **0.74** | **0.74** | 0.61 | 0.71 | 62 |
| Dead | 0.00 | 0.00 | 0.00 | 0.00 | 0.00 | 0.00 | 0.00 | 0.00 | 0.00 | 0.00 | 0.00 | 0.00 | 1 |

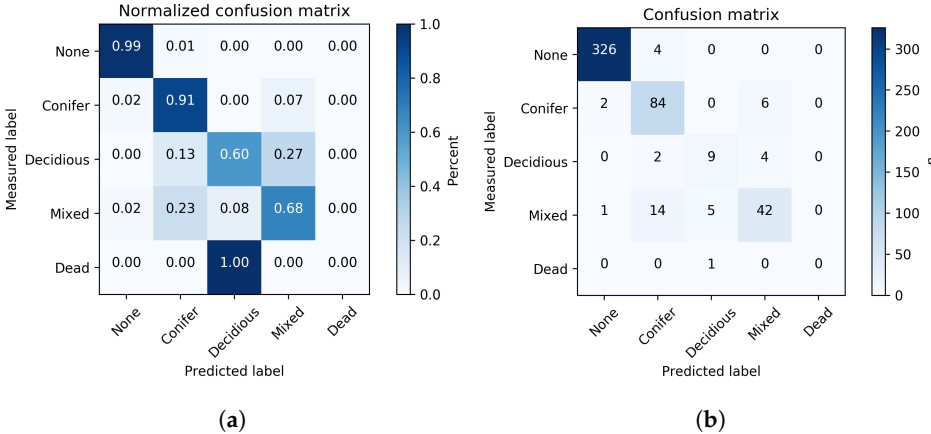

(**a**)                    (**b**)

**Figure 6.** Chimera ensemble classification task confusion matrix after 90 epochs of training, (**a**) normalized accuracy; (**b**) absolute accuracy counts. Low accuracy rate for 'dead' class relates to low number of total available training/validation samples. Increased 'dead' class samples could adjust for this problem.

### 3.1.2. Regression Task Diagnostics

The CE presented a strong ability to predict forest structure metrics, with all metrics reporting $R^2$'s above 0.8 on the independent test data ($n = 500$) (Figure 7). Scores for the single best individual model (BIM) and ensemble were similar, but with the ensemble consistently outperforming the BIM at all forest

structural metrics. Canopy cover was the best performing metric ($R^2 = 0.89$, RMSE = 8.01 percent), followed by basal area ($R^2 = 0.87$, RMSE = 25.88 ft$^2$/ac), AGB ($R^2 = 0.84$, RMSE = 37.28 Mg/ha), and finally QMD being the lowest ($R^2 = 0.81$, RMSE = 3.74 inches) (Table 4). This was expected, as elements of forest structure such as QMD underneath the canopy are hidden and can be difficult to estimate based on optically sensed spatial feature detection alone, whereas canopy cover features can often be recognized based on imagery alone.

Similarly, in spatially explicit prediction, plots demonstrated the ability for the model to be able to distinguish dense forest canopy cover and provided representative estimates of large and small diameter tree stands reflected with low QMD and basal area estimates (Figures 8 and 9)

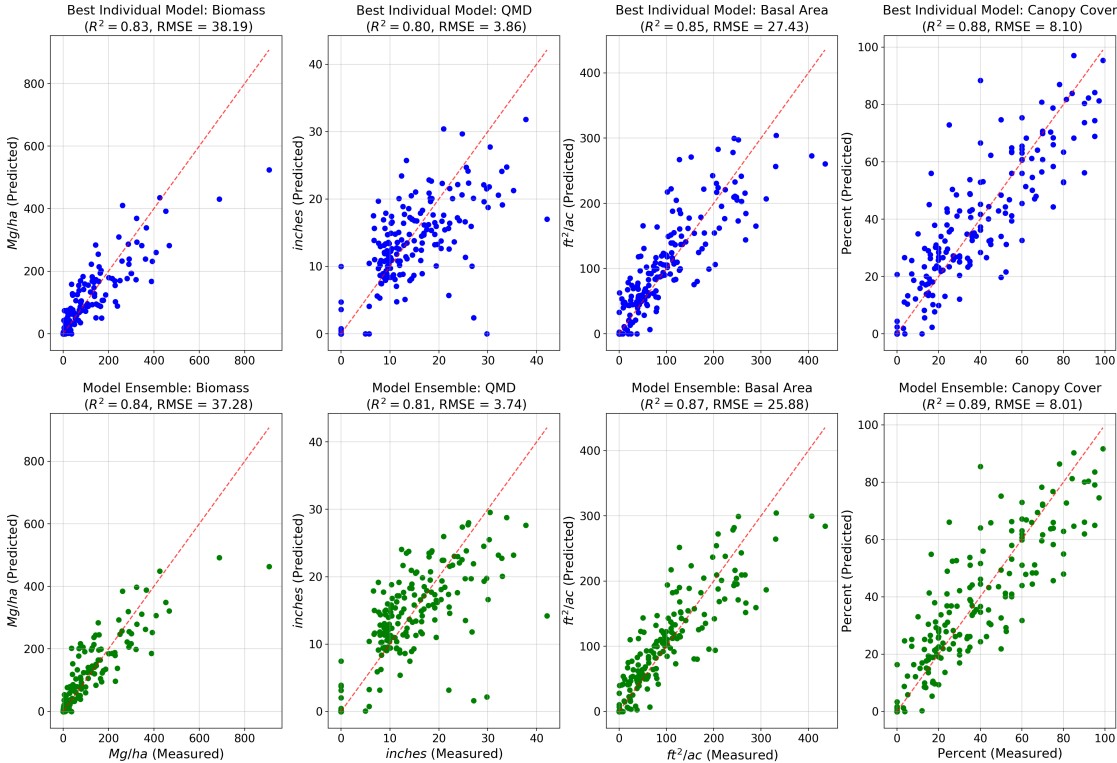

**Figure 7.** Regression diagnostics, showing true (*x*-axis) versus estimated (*y*-axis) for each regression variable on the $n = 500$ independent test dataset. The best performing individual model (**top**) and ensembled model (**bottom**) are shown.

**Table 4.** Regression accuracy metrics for the BIM and CE used in this study to predict forest structure metrics (above ground biomass (AGB), quadratic mean diameter (QMD), basal area, and canopy cover) from FIA plot data, compared with RF and SVM methods. All statistics reported here are for the training-withheld test set of 500 plots. Bold text highlights the best performing model in each category.

| Metric ($R^2$) | CE | BIM | RF | SVM |
|---|---|---|---|---|
| AGB | **0.84** | 0.83 | 0.76 | 0.68 |
| QMD | **0.81** | 0.80 | 0.76 | 0.69 |
| Basal Area | **0.87** | 0.85 | 0.78 | 0.79 |
| Canopy Cover | **0.89** | 0.88 | 0.84 | 0.84 |
| **Metric (*RMSE*)** | **CE** | **BIM** | **RF** | **SVM** |
| AGB (Mg/ha) | **37.28** | 38.19 | 45.71 | 52.11 |
| QMD (in) | **3.74** | 3.86 | 4.19 | 4.81 |
| Basal Area (ft$^2$/ac) | **25.88** | 27.43 | 33.16 | 32.73 |
| Canopy Cover (%) | **8.01** | 8.10 | 9.64 | 9.68 |

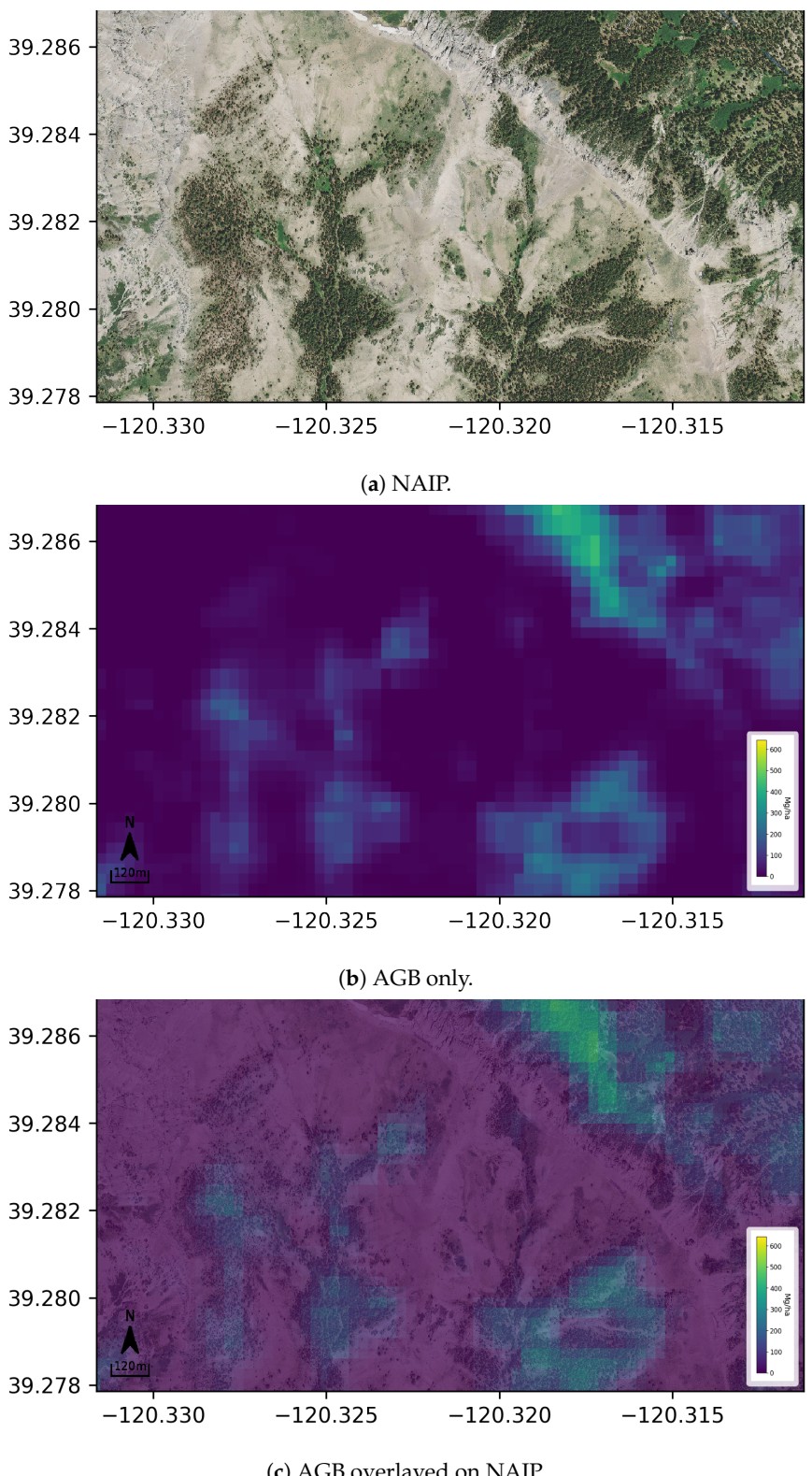

**Figure 8.** Example of AGB overlay plot on input NAIP aerial imagery. (**a**) NAIP aerial imagery alone; (**b**) AGB estimate; (**c**) AGB estimate overlaid on NAIP. These images aid in verification of the Chimera ensemble's ability to interpret image textures similarly to humans in distinguishing high and low density forests.

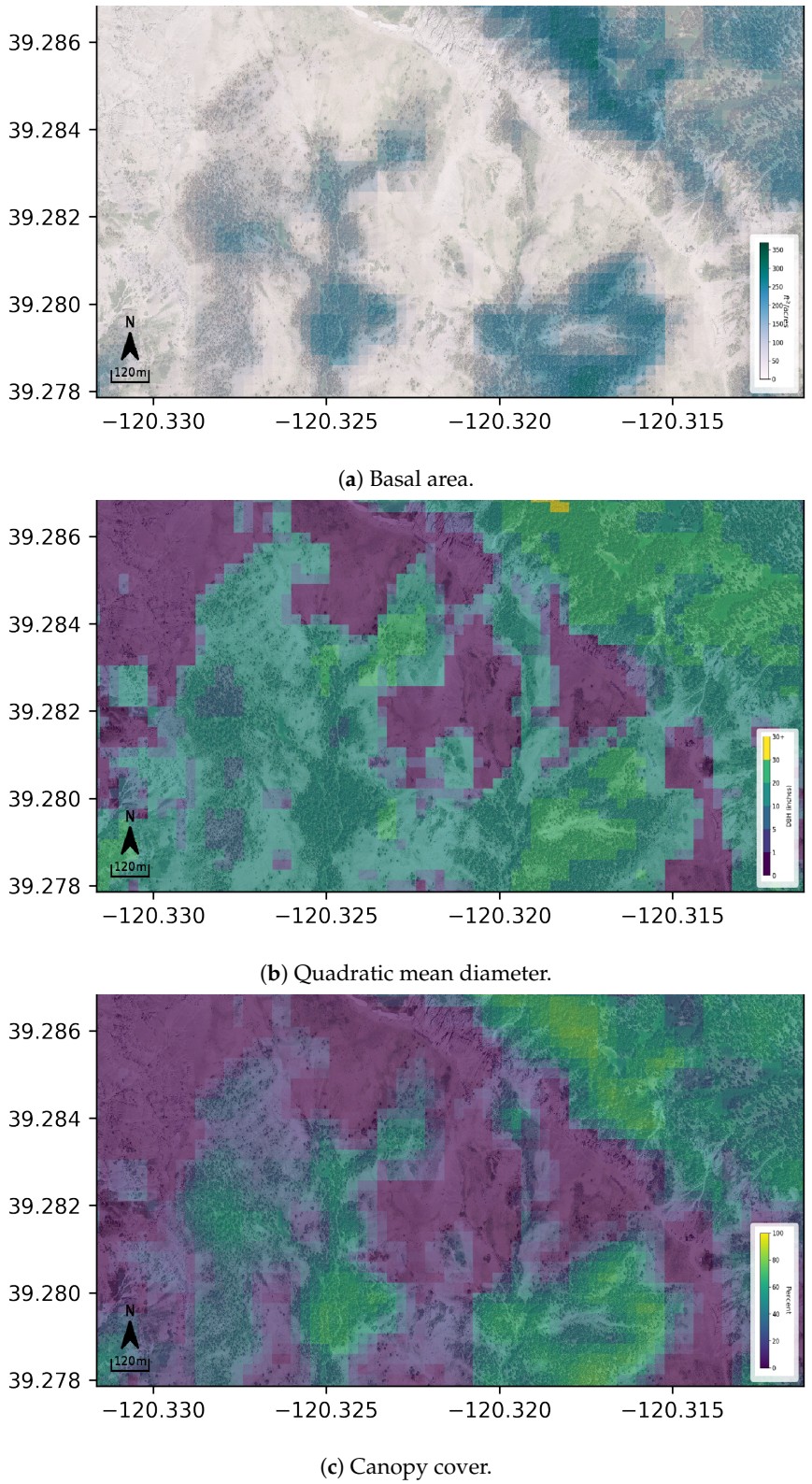

(**a**) Basal area.

(**b**) Quadratic mean diameter.

(**c**) Canopy cover.

**Figure 9.** Spatially explicit forest structure metric prediction plots overlain on NAIP (**a**) basal area; (**b**) quadratic mean diameter (QMD); (**c**) canopy cover, in the Lake Tahoe region. Chimera ensemble predictions of forest structure metrics are presented at 30-m resolution and can distinguish heterogeneous forested and non-forested cover.

*3.2. Model Comparison Experiment*

We explored two versions of SVM and RF models with a single-pixel seven-channel representation of Landsat data, and a 4 pixel × 4 pixel × seven-channel input. We found the single pixel variation resulted in the highest accuracy. RF outperformed SVM in all response variable estimates except basal area. The largest improvements by the Chimera ensemble were observed for AGB and basal area with a 0.08 increase in $R^2$'s respectively when compared to RF AGB and SVM basal area. Both the BIM $k$-fold and CE were superior to SVM and RF in all classification tasks except dead (Table 3). Highest classification difference was between the 'deciduous' type, with CE having a 0.14 increase in F-score compared to RF. CE also outperformed RF and SVM in regression in all forest structure metrics with the biggest improvements in basal area and AGB (Table 4). Post-hoc comparisons of saturation levels for estimated AGB values were modeled using a saturating exponential function [74,75] defined by, $\gamma(h) = c_0 + c_1(1 - e^{\frac{-3h}{a}})$, to identify the data saturation asymptote ($c_1$) for AGB prediction. This comparison reported Chimera reaching AGB data saturation much later (416.7 Mg/ha) than RF (386.3 Mg/ha) and SVM (245.0 Mg/ha), which agrees with its ability to perform better at estimating AGB (Figure A2).

## 4. Discussion

This research describes an effort to measure the ability of a novel deep learning approach for estimation of forest structure and classification simultaneously based on a fusion of multi-resolution and temporal data. There has been extensive research on estimating forest structure given the development of new modeling methodologies and the availability of free remote sensing data [6,7,76]. Implementations include utilization of various combinations of models and datasets to address forest structure estimates and classification [35,59,77–80]. Blackard et al. [11], performed a national scale effort to model AGB and forest classification through a tree-based algorithm approach with impressive results for AGB estimation utilizing three eight-day composited MODIS images, classified land cover information, and climate/topographic data. Three issues were cited in the study: (1) over-prediction of areas of small AGB and under-prediction of areas of large AGB due to reflectance saturation, (2) not capturing the full range in variability due to pixel size mismatch (250-m MODIS derived data) to FIA plot size, and (3) error in forest/non-forest classification due to FIA plot-based estimates pertaining to forest land use, which does not necessarily have trees on them, while satellite image-based estimates portraying forest land cover. Wilson et al. [78] made another step forward in this effort utilizing a phenological gradient nearest neighbor approach which integrating MODIS imagery across multiple time intervals, rather than a single aggregated time-step, and ancillary environmental data that includes topography and climate, to initially estimate forest basal area at the species level and later include AGB [8]. Our research examines a method to further include high resolution NAIP imagery in the suite of predictor variables, and utilize a unique RCNN architecture we call Chimera.

Previous work in utilizing more moderate spatial resolutions (30-m) remote sensing for estimating AGB had found that applications of machine learning methodologies more accurately estimated AGB than classical statistical methods, with an $R^2$ ranging from 0.54–0.78 [81–83]. Zolkos et al. [6] states that, although there are no explicit required accuracy levels for AGB estimation, a map for global AGB should not exceed errors of 50 Mg/ha at 1-ha resolution. This de facto standard has led to an increased interest in more advanced sensors to improve AGB estimation. For example, Zhang et al. [84] utilized Landsat and the Geoscience Laser Altimeter System (GLAS) space-borne platform in a large-scale modeling effort to produce gridded leaf-area index maps and canopy height maps of California, which derived AGB at a 30-m resolution across the entire state. Modeled uncertainties via their Monte Carlo methodology were in the range of 50–150 Mg/ha across the entire state, with denser forest in the Sierra Nevada Mountains closer to 100–150 Mg/ha. Chen et al. [85] was able to report within-sample statistics of $R^2 = 0.83$ and RMSE $= 72.2$ Mg/ha using a model based solely on aerial LiDAR for the Lake Tahoe region. This effort represents one of the best models for AGB estimation in the region. The Chimera ensemble displays the

ability to achieve similar performance in a comparable forested type landscape with high coefficient of determination ($R^2$) values (above 0.8) and low RMSE for all forest structure metrics.

As our first step at measuring the RCNN performance at the predictor level, we ran experiments comparing various permutations which included different combinations of the input NAIP, Landsat, and ANC data. We found that combining the full suite of data performed better overall than any of the input datatypes in isolation for forest structure estimation, specifically with the highest performance increases in AGB and basal area accuracy. The inclusion of high resolution information increased the data saturation level for estimates of AGB, where estimates reached saturation at 416.7 Mg/ha, rather than 354.0 Mg/ha, when only including the Landsat time-series and ancillary variables (Figure A2). The combination of NAIP and Landsat are found to have the best performance at the classification task overall versus Landsat alone, with an overall F1-score improvement of 0.08, which suggests that inclusion of high resolution imagery aids in AGB estimation and identification of forested and non-forested plots due to the additional textural information. This is in agreement with previous studies which demonstrated that including high resolution imagery increases a machine learning model's ability to distinguish forest composition types [86,87].

Our experiments with RF and SVM models using the same training and testing datasets further demonstrated how the RCNN method of spatial feature detection can improve forest structure estimation and classification. Due to the non-convolutional characteristics of RF and SVM, even if textural summarizing functions such as SDGL were included in the predictor datasets, the RCNN model was still able to achieve higher accuracy in both classification and regression tasks. The convolutional nature is important in the regression task, where structural features such as the size of a large or small canopy can be relevant for estimates of basal area and QMD. This characteristic contributed to the CE's increased performance versus RF, with an overall 17% reduction in RMSE for all forest structure metrics (Table 4). Additionally, because the Chimera model utilizes a recurrent layer in its structure for Landsat images, features that are identified in the twelve month series are seen as a set of orderly and continuing sequences, rather than a one-dimensional vector of independent values [88]. This allowed substantial improvement in distinguishing of the land use class of 'deciduous', with CE improving the F1-score by 0.26 and 0.14 compared to SVM and RF, respectively. This indicates that the Chimera model, with its ability to automatically identify relevant image features and incorporate information regarding spatial-temporal relationships between pixels in prediction, can efficiently and substantially improve forest structure estimation using the same input datasets.

Model ensembling/stacking was found to be an improvement over a single best fit Chimera model. Although only a modest improvement was found versus the best individual *k*-fold model with an increase of 0.025 in overall F1-score for classification, and regression improvements of overall $R^2$ of 0.0125, and reduction of overall RMSE by 3.1%. These findings were consistent with the literature that stacking multiple models based on a linear regression weighting, can reduce individual model biases and increase the robustness of prediction [89]. These biases could include image distortion, object occlusion, and inconsistent lighting being abundant in NAIP training data due to off-nadir image acquisition, and Landsat seven commonly containing gaps due to scanline errors and poor quality pixels. By fitting multiple models and predicting in an ensemble, we show that the Chimera ensemble is able to qualitatively estimate land cover type and forest structure well (Figures A3 and A5).

*Caveats Associated with the Training Data*

Although 27,966 FIA plots were available for the CA-NV study area, we removed a large proportion (>50%) of the samples after manual visual inspection, in which non-forest land-use classification defined by FIA did not correspond to the associated land cover in the NAIP image. Though time-consuming and laborious, this exercise was essential to prevent the model from learning incorrect image texture representations of the forest classes and structural attributes measured in situ [21]. The performance of Chimera can mainly be attributed to its access to those 9967 quality data plots. CNNs are the most data-hungry method of machine learning, primarily due to the sheer

number of parameters needed to fit the model, which in turn is based on the volume of training data. We also note that given the differences in FIA protocols over time and regions [61], some forest structure response attributes such as canopy cover were measured subject to variability (in situ versus remote sensed image interpretation), which could have downstream effects on the performance of CE (Figure A5). In terms of portability and robustness, it should be mentioned that our tests of CE are exclusively in the CA-NV region. Although a variety of US Western ecotypes, from temperate forests to semi-arid woodlands are represented in our analysis, our approach would not be applicable to make predictions of forest class or structure in rainforest ecotypes nor the hardwood forests of the Eastern US, unless CE were retrained on data from those specific forest types.

## 5. Conclusions and Future Research

We demonstrate the performance of a novel multi-task, multi-input recurrent convolutional neural network called the Chimera model for forest land use classification and forest structure estimation. We summarized the results of three major objectives as follows: (1) performance of the Chimera model was the highest with the full input dataset that included NAIP, Landsat time-series, and ancillary climate and topography data; (2) the Chimera model outperforms SVM and RF models with the same input data for all classification and regression tasks; and (3) ensembling of multiple Chimera models modestly increased predictive performance compared to a single best fit model alone. These results represent, to our knowledge, the first application of a RCNN with multi-input for improving estimations of forest structure.

The ability of the Chimera ensemble to distinguish between forested and non-forested land cover images within California and Nevada presents a new approach for generating a time-series of change detection for both afforestation and deforestation based on high-resolution imagery. Since our model requires only freely accessible data, we can feed new acquisitions of NAIP and Landsat scenes through the existing model to generate new predictions. This potential for continuous prediction is progress towards work similar to Wilson et al. [8] who developed a model for imputing forest carbon stock at a nationally continuous scale. Given access to the national USFS FIA dataset, widely considered the "gold standard" of field collected forest metric samples within the entire United States, one would have a large enough sample size to parameterize multiple robust RCNN models focused on a forest of interest. A future iteration of the Chimera ensemble trained on a new subset of USFS FIA encompassing a region outside of California and Nevada could provide state-of-the-art estimates of forest structure in many completely different ecotypes. Additionally, with integration of advanced satellite technology including the GEDI and NISAR missions [90,91], more opportunities to combine LiDAR or radar information with existing NAIP and Landsat data could generate still better estimates of forest structure.

Finally, we see this study as just one indication of the promising application of deep learning architectures to ecology and remote sensing. Future work could include multi-task learning applied to land-cover and land-use mapping problems such as monitoring of woody encroachment on wetlands, or measuring forest loss due to wildfire or disease outbreaks. In addition to forest and stand structural characteristics, future research could inform the identification and delineation of key habitat attributes for wildlife [92]. Wall-to-wall forest structural maps can also be used as inputs for fire modeling applications that depend on canopy cover, canopy height, canopy crown bulk density, and crown base height to determine fire behavior [93]. Integration of our model in a "near real-time" wildfire probability model could provide estimates of fire risk and serve as a tool for prioritizing locations for fire prevention treatments [94]. Predictive modeling applications (e.g., coastline vegetation, wetlands, grasslands, or shrublands) often require both classification (e.g., presence/absence, type) and regression (e.g., abundance/cover conditional on presence).

We have demonstrated that the Chimera architecture is capable of fusing various types of data (sensor, climate, and geophysical data, both time-varying and fixed) that are now commonplace in many ecological applications, and using the information they provide to improve predictive ability.

By taking advantage of distributed cloud computing, future development of a continuously integrated pipeline could extend this model towards automatically updated landscape or global forest metric estimates. RCNN architectures similar to Chimera present an opportunity to combine disparate data types and move towards global-to-local level ecosystem monitoring.

**Author Contributions:** Conceptualization, T.C., B.G.D., and L.J.Z.; methodology, T.C. and B.P.R.; software, T.C. and B.P.R.; validation, T.C. and B.P.R.; formal analysis, T.C. and B.P.R.; investigation, T.C. and B.P.R.; resources, T.C.; data curation, B.P.R.; writing—original draft preparation, T.C., B.P.R., B.G.D., and L.J.Z.; writing—review and editing, T.C., B.P.R., B.G.D., and L.J.Z.; visualization, T.C. and B.P.R.; supervision, T.C. and B.G.D.; project administration, T.C. and B.G.D.; funding acquisition, T.C., B.G.D., and L.J.Z.

**Funding:** This research was funded by the David H. Smith Conservation Research Fellowship administered by the Society for Conservation Biology and financially supported by the Cedar Tree Foundation. Additional support was provided by Living Forests and the University of California, Santa Cruz (on behalf of The Center for the Study of the Force Majeure, Joshua Harrison PI), and the Tahoe Truckee Community Foundation. This research was made possible through an agreement between Conservation Science Partners and the USDA FIA Program through the Regional FIA Spatial Data Services Contact under the terms of MOU 18-MU-11261979-015 and MOU 18-MU-11221638-060. Microsoft Azure Cloud computational support was provided through the Microsoft AI for Earth Grant. The APC was funded by Tahoe Truckee Community Foundation.

**Acknowledgments:** We thank Richard McCullough, Thomas Thompson, John M. Chase, Barry Ty Wilson, Jacob Strunk, Chris Toney, and other anonymous reviewers from the USDA Forest Service for their valuable feedback.

**Conflicts of Interest:** The authors declare no conflict of interest. The funders had no role in the design of the study; in the collection, analyses, or interpretation of data; in the writing of the manuscript, or in the decision to publish the results.

## Abbreviations

The following abbreviations are used in this manuscript:

| | |
|---|---|
| AGB | Above ground biomass |
| ANC | Ancillary climate and terrain data |
| BIM | Best individual chimera model |
| CA | California |
| CE | Chimera ensemble |
| CNN | Convolutional neural network |
| DEM | Digital elevation model |
| GEDI | Global Ecosystem Dynamics Investigation LiDAR |
| GEE | Google Earth Engine |
| GLAS | Geoscience laser altimeter system |
| GLCM | Gray level co-occurrence matrix |
| FIA | Forest inventory and analysis |
| LS | Landsat |
| LSTM | Long-short term memory cell |
| MODIS | Moderate resolution imaging spectroradiometer |
| MTL | Multi-task learning |
| NASA | National Aeronautics and Space Administration |
| NISAR | NASA-ISRO Synthetic Aperture Radar |
| NAIP | National Agriculture Inventory Program |
| NED | National elevation dataset |
| NV | Nevada |
| PRISM | Parameter-elevation regressions on independent slopes model |
| QA | Quality assessment |
| QMD | Quadratic mean diameter |
| RCNN | Recurrent convolutional neural network |
| ReLU | Rectified linear units |
| RF | Random forest |
| RGB | Red-green-blue |
| RMSE | Root mean squared error |

SD          Standard deviation
SDGL        Standard deviation of gray levels
SVM         Support vector machine
TOA         Top Of atmosphere
USDA        United States Department of Agriculture
USGS        United States Geological Survey
YOLO        You only look once

## Appendix A. Additional Figures

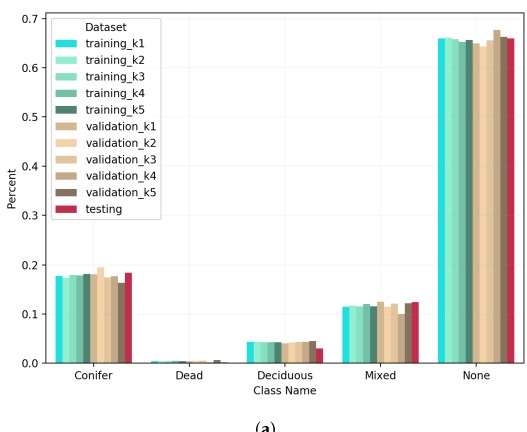

(**a**)

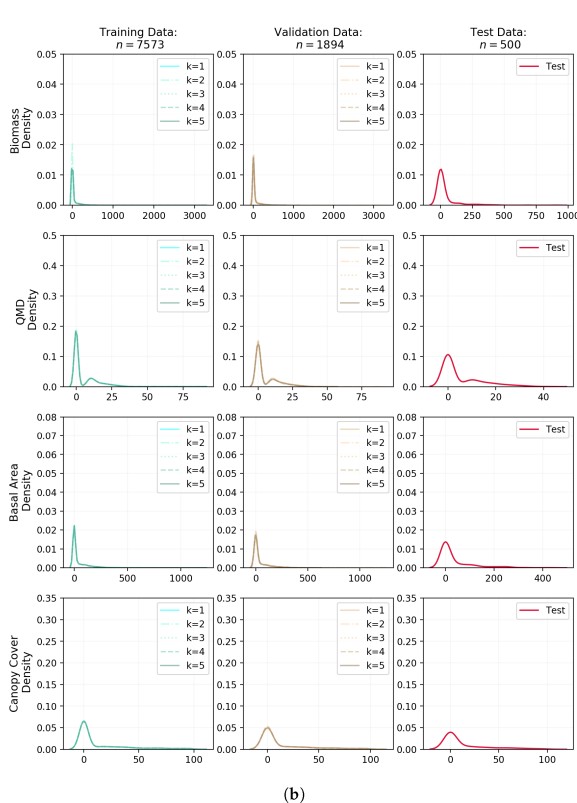

(**b**)

**Figure A1.** Distribution plots of training and validation *k*-fold subsets, and testing data demonstrate similar (**a**) percentages of each response variable type of land cover class and (**b**) densities plots of each forest structure metric.

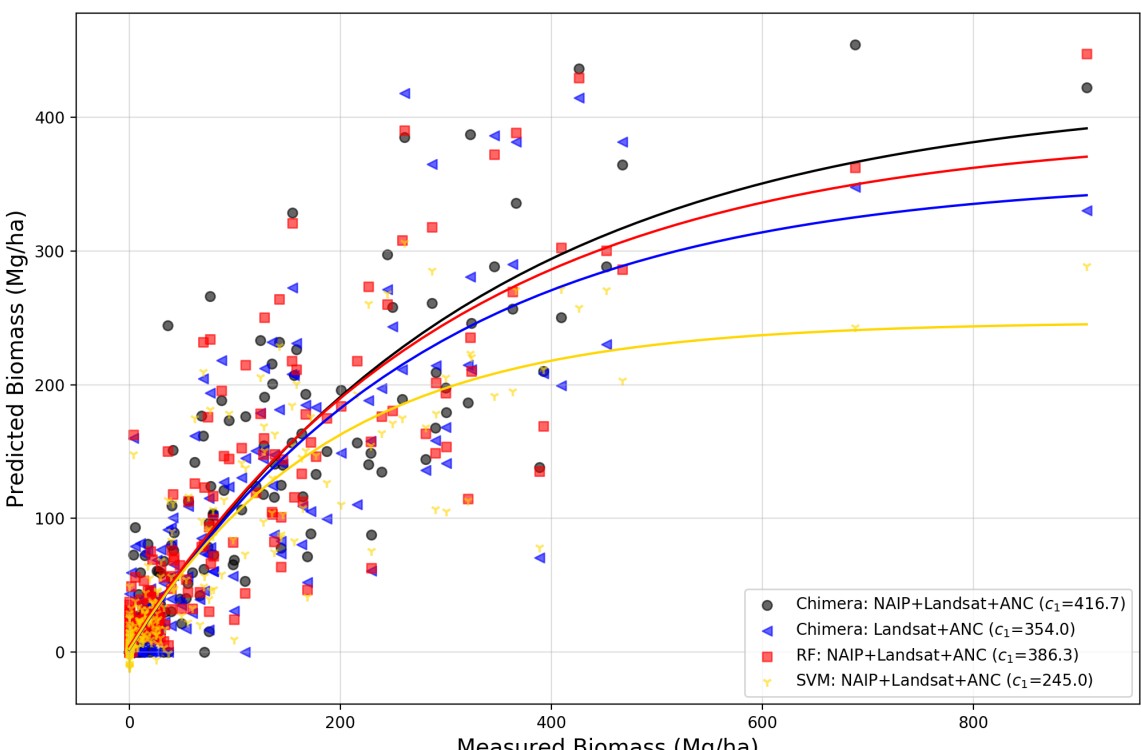

**Figure A2.** Plots of predicted AGB versus measured AGB within test set comparison amongst different input combinations of the Chimera model and SVM and RF. The NAIP + ANC and full NAIP + Landsat + ANC displayed the highest predictive values prior to reaching saturation.

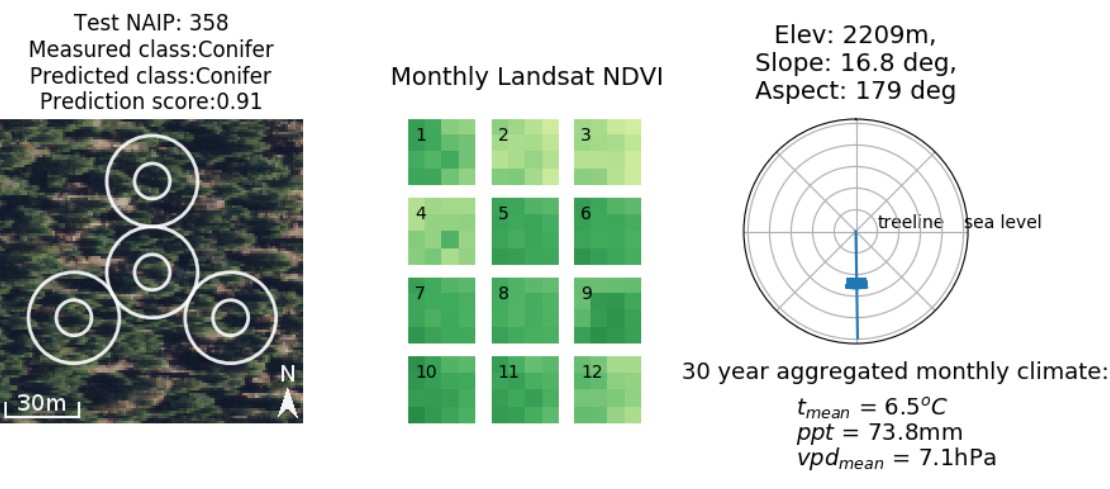

| | Biomass (*Mg/ha*) | QMD (*inches*) | Basal Area (*ft²/ac*) | Canopy Cover (%) |
|---|---|---|---|---|
| Measured | 137.99 | 16.89 | 172.98 | 60.00 |
| Predicted | 142.39 | 16.96 | 186.60 | 51.78 |

(**a**)

**Figure A3.** *Cont.*

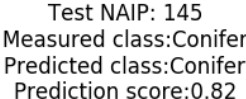

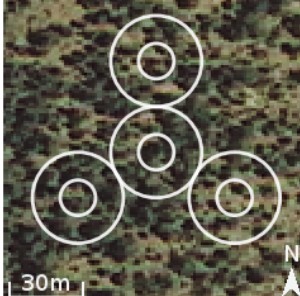

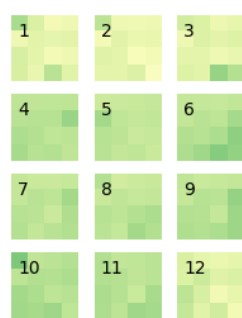

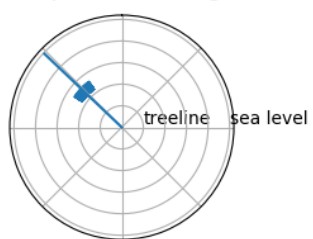

| | Biomass (*Mg/ha*) | QMD (*inches*) | Basal Area (*ft²/ac*) | Canopy Cover (%) |
|---|---|---|---|---|
| Measured | 26.53 | 9.65 | 99.34 | 35.00 |
| Predicted | 32.77 | 11.55 | 126.99 | 39.66 |

(**b**)

**Figure A3.** Visual representations of a sample (not all inputs presented here) used for training the Chimera ensemble. White outlined area represents plot regions for FIA field measurement and total image represents 120-m × 120-m area at 1-m resolution. Fourteen total climate parameters, three terrain characteristics, and all Landsat bands except thermal bands were used for training, as well as RGB NAIP. Landsat images represent 4 × 4 pixels at 30-m resolution. In addition to AGB of live standing trees >= 1 inch (2.54 cm) in diameter, other prediction values generated from FIA samples included QMD, percent canopy cover, basal area, and forest plot classification. Examples of 'conifer' class plots, demonstrate the ability to identify textural features of both (**a**) heavy timber and (**b**) woodland conifer forests.

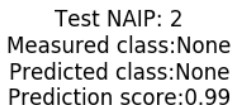

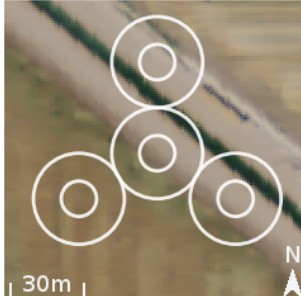

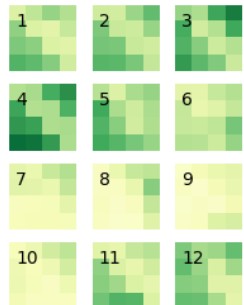

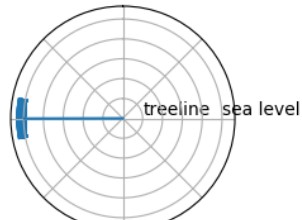

| | Biomass (*Mg/ha*) | QMD (*inches*) | Basal Area (*ft²/ac*) | Canopy Cover (%) |
|---|---|---|---|---|
| Measured | 0.00 | 0.00 | 0.00 | 0.00 |
| Predicted | 0.00 | 0.00 | 0.00 | 0.00 |

(**a**)

**Figure A4.** *Cont.*

Test NAIP: 404
Measured class:None
Predicted class:None
Prediction score:0.98

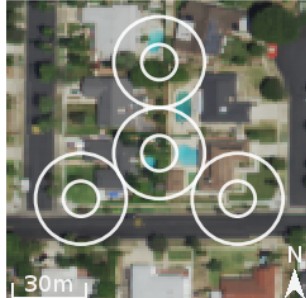

Monthly Landsat NDVI

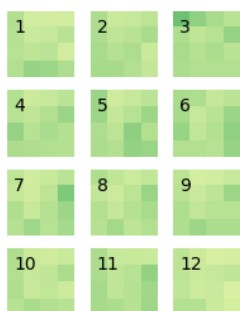

Elev: 48m,
Slope: 1.5 deg,
Aspect: 343 deg

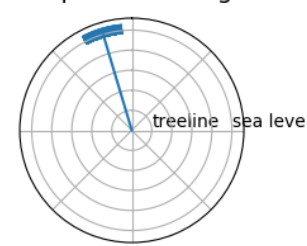

treeline    sea level

30 year aggregated monthly climate:

$$t_{mean} = 18.5^{o}C$$
$$ppt = 25.7mm$$
$$vpd_{mean} = 9.9hPa$$

|  | Biomass (*Mg/ha*) | QMD (*inches*) | Basal Area (*ft²/ac*) | Canopy Cover (%) |
|---|---|---|---|---|
| Measured | 0.00 | 0.00 | 0.00 | 0.00 |
| Predicted | 0.00 | 0.00 | 0.00 | 0.00 |

(**b**)

**Figure A4.** Examples of 'none' class (non-forested) plots. Chimera ensemble can distinguish features from both (**a**) roads/agriculture and (**b**) urban featured areas from forested areas.

Test NAIP: 62
Measured class:Mixed
Predicted class:Mixed
Prediction score:0.70

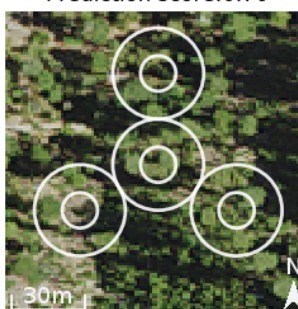

Monthly Landsat NDVI

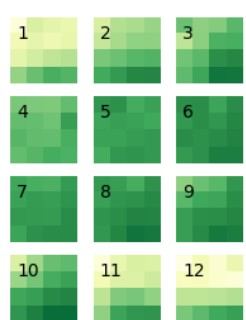

Elev: 1264m,
Slope: 28.0 deg,
Aspect: 14 deg

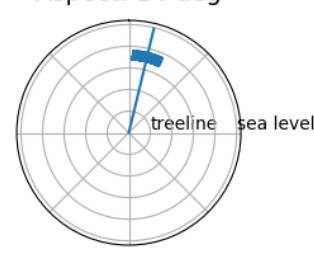

treeline    sea level

30 year aggregated monthly climate:

$$t_{mean} = 11.0^{o}C$$
$$ppt = 150.0mm$$
$$vpd_{mean} = 8.0hPa$$

|  | Biomass (*Mg/ha*) | QMD (*inches*) | Basal Area (*ft²/ac*) | Canopy Cover (%) |
|---|---|---|---|---|
| Measured | 187.20 | 20.45 | 111.94 | 55.00 |
| Predicted | 134.25 | 17.24 | 113.66 | 52.59 |

(**a**)

**Figure A5.** *Cont.*

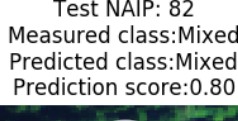

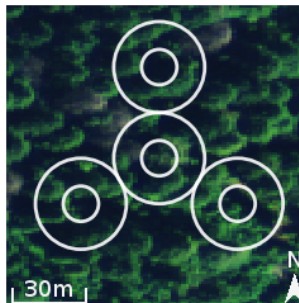

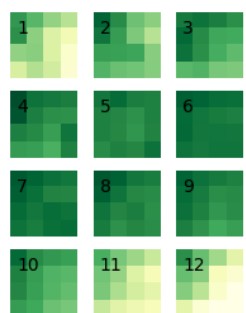

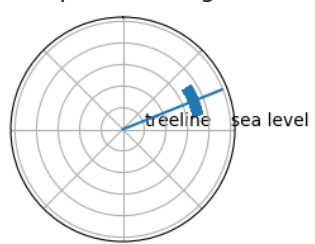

| | Biomass (*Mg/ha*) | QMD (*inches*) | Basal Area (*ft²/ac*) | Canopy Cover (%) |
|---|---|---|---|---|
| Measured | 426.00 | 25.93 | 243.02 | 40.00 |
| Predicted | 447.53 | 26.86 | 281.49 | 83.99 |

(**b**)

**Figure A5.** Example of 'mixed' class plots, where model identifies semi-green saturation from the Landsat temporal signal. (**a**) Case of likely-accurate underestimation of AGB and canopy cover due to sparseness of vegetation in unsampled region of image. (**b**) Challenging case of over estimation for forest canopy cover in prediction, which are larger than in situ plot measurement from heavily shadowed NAIP visual signal.

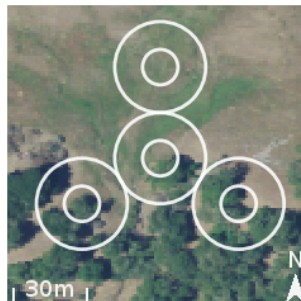

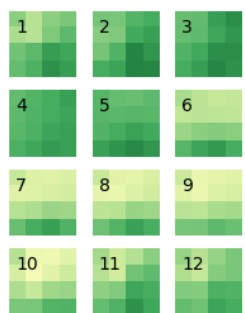

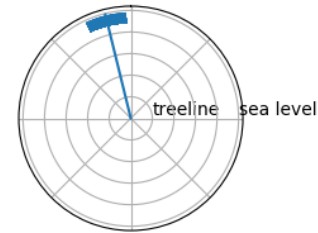

| | Biomass (*Mg/ha*) | QMD (*inches*) | Basal Area (*ft²/ac*) | Canopy Cover (%) |
|---|---|---|---|---|
| Measured | 99.13 | 18.18 | 79.49 | 52.33 |
| Predicted | 58.99 | 18.90 | 48.56 | 28.57 |

(**a**)

**Figure A6.** *Cont.*

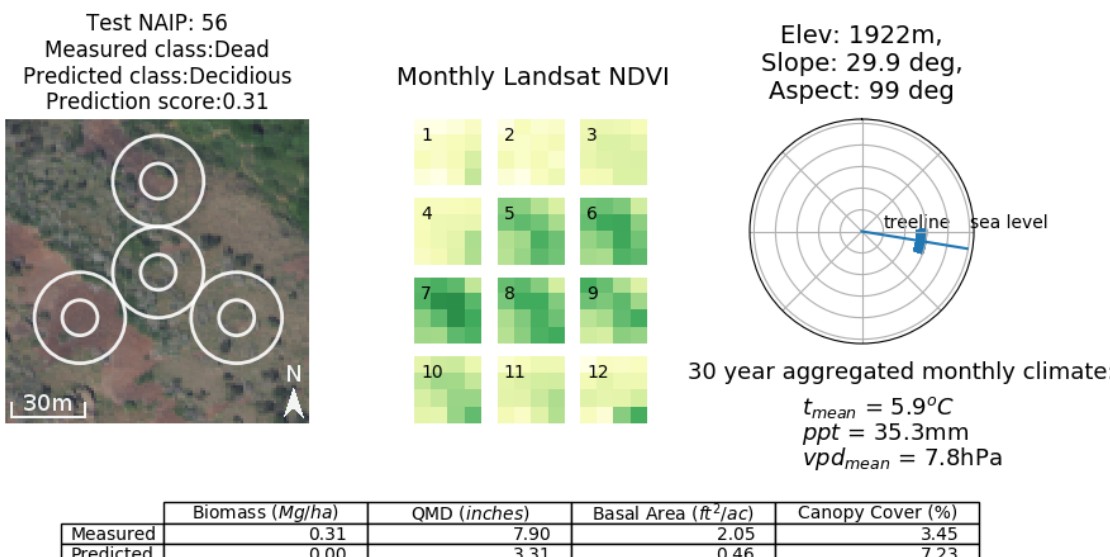

(**b**)

**Figure A6.** Example of challenging 'deciduous' and 'dead' stands. (**a**) 'deciduous' example was half populated with trees within the NAIP image sample. (**b**) 'dead' case reveals a mixed Landsat signal, leading to a misclassification of 'deciduous' with low prediction probability.

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
