# Peer review of "Chimera: A Multi-Task Recurrent Convolutional Neural Network for Forest Classification and Structural Estimation"

_remotesensing, doi:10.3390/rs11070768_

Round 1

Reviewer 1 Report

It is important to classify forest vegetation types and their metrics with high accuracy in the current global climate change scenario. The present study introduces a multi-task recurrent convolutional neural network model called “Chimera” for forest classification and its metrics. The model uses the fusion of several remotely sensed datasets at different resolutions for forest classification and calculating metrics using FIA plots for training and validation. Also, the results of the model are compared with other techniques such as random forests and SVM’s and the results show that Chimera ensembles produce more accurate results for classification and metrics than other techniques. The study is well designed and I only have a few minor comments regarding the manuscript.

- I felt the abstract to be confusing and not representing completely what the authors tried to do. It was clear in the introduction what the authors intended to do and hence the authors should rewrite the whole abstract to increase the comprehensibility. I felt Chimera to be an already existing model while reading the abstract (not a new model you were trying to introduce).

L 3- Is “density” proper usage?

L 10- expand AGB here.

LL 109 – 111- It was confusing to read this sentence. Rewrite for better comprehensibility. Do the same for the following comments.

LL 113-115-Rewrite

LL 121 – 122 – Rewrite

LL 155-156- Rewrite

-It would have been better if you could also give the resolution of different datasets in m in methods. Here, some places resolution is given in m, some places in arc minutes and arc seconds.

-Give details about data used for validation of each class in Table 1.

Author Response

Thank you for your review and comments. Please see the attached document where we have addressed each comment point-by-point found in bold font.

Reviewer 2 Report

MANUSCRIPT REVIEW

MDPI Remote Sensing 470666 - Chang et al. 2019 - Chimera a multi-task recurrent convolutional neural network for forest classification and structural estimation

Summary of review (detailed point by point comments follow):

The manuscript is well written and interesting for the readers of MDPI Remote Sensing, with a highly relevant and novel method based on ensemble of RCNNs performing simultaneous classification and regression of multiple forest variables.

The introduction is overall quite good, but logical transitions between some paragraphs could be improved.

Some claims about CNNs or deep learning methods are either not exact or exaggerated and should be rephrased.

The low proportion of samples used in test dataset (5%) creates an issue with the least frequent class, that could be partially mitigated by using more test samples or sampling by class importance. It would be interesting how the models perform with slightly less training samples but more test samples, this should be an easy experiment to run.

The list of references is thorough, however some references chosen as machine learning reviews are dated, better alternative exist in the field of remote sensing.

Some figures should follow MDPI guidelines.

Experiments are significant and results adequate for publication. The discussion section is thorough.

Therefore I recommend a minor revision. On the next pages of this review are suggestions for changes (→), P = page, L = line

Main comments

·         L 41-42 : authors suggest it is the sub-metric resolution of LiDAR sensor that allows a high level of accuracy in AGB estimation. If that was the case, similar AGB accuracy will be obtained with sub-metric optical satellite images. I would argue the ability to estimate tree height is what gives LiDAR the edge for AGB estimation.

·         L59 : this sentence mentions “newer machine learning methods, specifically deep learning”, and ends on L63 with references [23-27] to mostly older (1997, 1999 or early 2000s), non deep learning works. Please make the claim and references match. One could add for example Zhu et al., 2017, among others. One could also add Lopez-Serrano et al., 2016 for a comparison on machine learning methods (not deep) applied to Landsat data for biomass estimation.

·         L65 “The ability of CNNs to recognize patterns in multi-dimensional domains that have both time and space components” : agreed on spatial patterns as shown with the many successes of CNNs in computer vision field. But CNNs are not naturally good with temporal domain - otherwise RNN, RCNN and LSTM would not have the edge over CNN in change detection or time series analysis.

·         L68-73 : authors claim CNNs identify texture features automatically better than hand-crafted texture features. CNNs extract spatial features very well in general, but are not specialised in texture extraction therefore can still benefit from adding hand-crafted texture feature. It has been demonstrated recently on several optical remote sensing benchmark datasets that integrating a simple explicit texture measure as input to a CNN network improves classification accuracy on all classes including forest (Anwer et al., 2018).

·         L77 - 78 : the logical transition between those two paragraphs is a bit weaker than the rest of introduction. Authors could better explain why ensemble approaches are needed, and better justify why a unified approach to perform classification and regression tasks simultaneously is needed as well.

·         About the random sampling scheme and class imbalance :

L198 - 200 : out of 9 967 samples in total, 500 (about 5%) are used in test dataset. This is a low percentage. Also the training, validation and test sets are selected randomly, which creates an issue for one class (‘dead trees’) that is heavily under-represented in the test set and therefore results cannot be evaluated on this class. If there are not enough samples for a proper accuracy assessment of this class, it should be dropped entirely or merged with another class. Or ideally more samples of this class should be collected to allow both training and fair evaluation of the models. It would be interesting to run additional experiments with 10% or 20% of samples in test set.

L276 in results, the ‘dead’ class only has one sample in test set. According to Table 1 there are 44 samples in ‘dead’ class. 5 % of samples went to test set in random sampling, so we could expect 2.5 maybe 3 samples of this class in test set. Which shows why random sampling should be weighted by class relative importance to allow for representativeness of training set vs test set. Was random selection stratified by classes or completely random ? Fig A1 seems to suggest that class proportions are roughly the same for training and validation sets.

·         L 306-324 : these reference works in Discussion utilize MODIS data which is much coarser than Landsat data. Are there no other works on AGB estimation with Landsat data, that would be more relevant to discuss and compare to the proposed approach ? I suggest to remove this paragraph and replace with more relevant comparison.

·         L330-332 : here the authors compare their work with a study in Lake Tahoe. Are there other works in California to compare the proposed work with ?

Minor comments

·         The authors use the term “high spatial resolution” for sensors like Quickbird or airborne LiDAR, I would rather use “very high spatial resolution” for sub-metric resolution.

·         L108 - 109 : what are the “four different open-source remote sensing datasets available on Google Earth Engine” that are used in this study ? NAIP, Landsat, ? and ? : please name them in this sentence, or the reader is left to wonder…

Syntax / typos / editing

·         In the whole article, RMSE values are giving without the corresponding unit. Please consider adding the relevant units, e.g. m2/ha for RMSE of basal area, m for RMSE of QMD etc…

·         lidar should be written “LiDAR” (occurs several times in the manuscript)

·         L10 : explain acronym AGB here (1st time used) and add ‘(QMD)’ after ‘diameter’

·         L12 “unfuzzed” : not clear… did you mean “not fused”, or something else ?

Figures and tables

·         Figures 1, 7 and 8: according to MDPI Remote sensing publishing guidelines, all figures containing satellite images must include coordinates (bounding box around the image), a scalebar indicating distances, and a north arrow.

·         Left-hand side of Figures A3, A4, A5 and A6 : same remark as for other figures

·         Figure 5 : switch captions of a) and b) - the confusion matrix on right-hand side (b) is for absolute values, not normalized

·         Caption of Figure 7 : “Spatially explicit AB estimate plots match expectations from human expert visual inspection of NAIP. This aids in verification of the Chimera ensemble’s ability to interpret image textures similarly to humans in distinguishing high and low density forest” : in order to back up this claim, one would need a more rigorous experimental protocol, involving more images in several sites, several human experts, a blind evaluation process, statistical samples and confidence intervals. Therefore this is a bit exaggerated claim, that needs rephrasing.

·         Table 2 : bolding the best result on each row (not only overall) would make the table even easier and faster to read.

·         Tables 3 and 4 : I suggest to switch columns CE and BIM : CE is the proposed ensemble method, others are baseline non-ensemble methods and should be next to each other in the table.

References

Here is the list of references suggested to be added in the manuscript.

·         X. X. Zhu et al., 2017. Deep Learning in Remote Sensing: A Comprehensive Review and List of Resources," in IEEE Geoscience and Remote Sensing Magazine, vol. 5, no. 4, pp. 8-36, Dec. 2017. https://doi.org/10.1109/MGRS.2017.2762307

·         López-Serrano et al., 2016. A Comparison of Machine Learning Techniques Applied to Landsat-5 TM Spectral Data for Biomass Estimation, Canadian Journal of Remote Sensing, 42:6, 690-705, DOI: https://doi.org/10.1080/07038992.2016.1217485

·         Anwer, R.M. et al., 2018. Binary patterns encoded convolutional neural networks for texture recognition and remote sensing scene classification. ISPRS Journal of Photogrammetry and Remote Sensing, Volume 138, 2018, Pages 74-85, https://doi.org/10.1016/j.isprsjprs.2018.01.023

·         Interdonato, et al., 2019. DuPLO: A DUal view Point deep Learning architecture for time series classificatiOn, ISPRS Journal of Photogrammetry and Remote Sensing, Volume 149, 2019, Pages 91-104,

https://doi.org/10.1016/j.isprsjprs.2019.01.011

Author Response

(The authors gave the same response as above.)

Reviewer 3 Report

The submitted paper addresses the ensemble of multi-task recurrent CNN varying resolution, freely available aerial and satellite imagery as well as relevant environmental factors (e.g., climate, terrain) to simultaneously classify five forest cover types (‘conifer’, ‘deciduous’, ‘mixed’, ‘dead’, ‘none’ (non-forest)) and to estimate four continuous forest structure metrics (AGB, quadratic mean diameter, basal area, canopy cover) from a representative area.

The topic is of high scientific relevance and is well within the scope of Remote Sensing Journal. However, manuscript could benefit from a technical review and proofreading (for example, there is no need to be repetitive and cite the value 0.92 several times on L14). A carefully check must also be performed since not all methodical procedures are properly described (e.g. F1-score: how to calculate and what does it means? Among others).

All database, including remotely sensed data and field measurements must be better described. Adding a flowchart of the major activities would be a good idea to enhance the paper.

I recommend to better explain better important issues that are
not sufficiently addressed in the current version of the manuscript: (i)
portability/robustness of the proposed method, and (ii) the implicitly
assumed relation between biophysical data and remotely sensed data. Please consider that the introduction section should provide a kind of state-of-the-art.

In my opinion, it is mandatory to complement/enhance Table 1 with a summarizing of the experimental data. Besides average and standard deviation, please add min and max values as well. Please add the as varying dot sizes the spatial variability/homogeneity of the
variables. If this spatial variability was not measured, please
include at least a verbal description. Similarly, are the ids of remote sensing data (e.g. path/raw of the Landsat data) and their timeframe acquisition. Saturation aspects are also not properly described.

Results in form of maps containing the spatial distribution of the classes and biophysical properties for the entire area (rather than only subsets that we even don’t know where they are located) could be presented as well.

Author Response

(The authors gave the same response as above.)

Round 2

Reviewer 3 Report

The authors presented reasonable concerns on their justifications. The addition of new figures makes the paper now more attractive for the readers. The concerns were properly performed in the revised paper. I am happy with the status and quality of the present version of the manuscript. Therefore, I am now recommending it for publication. Thank you for the opportunity to review this interesting research.